# Stable clonal contribution of lineage-restricted stem cells to human hematopoiesis

Tetsuichi Yoshizato [1] ✉, Christer Nilsson [1,2], Francesca Grasso[1],
Kari Högstrand[1], Stefania Mazzi [1], Axel Winroth [1], Madeleine Lehander[1],
Indira Barbosa [1], Gunilla Waldin[1], Teresa Mortera-Blanco[1], Monika Jansson[1],
Mikaela Hillberg Widfeldt[1], Affaf Aliouat[3], Margs S. Brennan [1],
Ellen Markljung[4], Amy Hillen [4], Edwin Chari[1], Eva Hellström-Lindberg [1,2],
Warren W. Kretzschmar [1], Petter S. Woll [1] & Sten Eirik W. Jacobsen [1,2,3,4] ✉

Dynamic steady-state lineage contribution of human hematopoietic stem cell (HSC) clones needs to be assessed over time. However, clonal contribution of HSCs has only been investigated at single time points and without assessing the critical erythroid and platelet lineages. Here we screened for somatic mutations in healthy aged individuals, identifying expanded HSC clones accessible for lineage tracing of all major blood cell lineages. In addition to HSC clones with balanced contribution to all lineages, we identified clones with all myeloid lineages but no or few B and T lymphocytes or all myeloid lineages and B cells but no T cells. No other lineage restriction patterns were reproducibly observed. Retrospective phylogenetic inferences uncovered a 'hierarchical' pattern of descendant subclones more lineage biased than their ancestral clone and a more common 'stable' pattern with descendant subclones showing highly concordant lineage contributions with their ancestral clone, despite decades of separation. Prospective lineage tracing confirmed remarkable stability over years of HSC clones with distinct lineage replenishment patterns.

Multipotent self-renewing HSCs, possessing the potential to replenish all mature blood cell lineages, are critical for safeguarding lifelong replenishment of millions of blood cells every second in steady-state hematopoiesis and in response to hematopoietic challenges[1,2]. Our knowledge of the functional properties and roles of mammalian HSCs is largely based on studies in mice[3] due to limitations of available platforms for assessing human HSC function[4]. Studies of steady-state replenishment of different critical blood cell lineages by human HSC clones have been hampered by an estimated 50,000–200,000 HSCs

actively contributing to hematopoiesis both in young and aged adults[5,6]. The high number of HSCs also makes it challenging to lineage trace HSC clones emerging in steady-state hematopoiesis in young adult mice[3,7–9]. Moreover, steady-state lineage contribution from individual HSC clones has only been investigated at single time points in mice and in humans[7,8,10–14].

In vivo fate mapping of single HSCs from young mice transplanted into myeloablated congenic recipients with a minimal number of genetically distinguishable HSCs competing for lineage replenishment has

[1]Department of Medicine Huddinge, Center for Hematology and Regenerative Medicine, Karolinska Institutet, Karolinska University Hospital Huddinge, Stockholm, Sweden. [2]Division of Hematology, Department of Medicine, Karolinska University Hospital Huddinge, Stockholm, Sweden. [3]Haematopoietic Stem Cell Biology Laboratory and MRC Molecular Haematology Unit, MRC Weatherall Institute of Molecular Medicine, University of Oxford, Oxford, UK. [4]Department of Cell and Molecular Biology, Karolinska Institutet, Stockholm, Sweden. ✉e-mail: tetsuichi.yoshizato@ki.se; sten.eirik.jacobsen@ki.se

established the existence of distinct HSC subsets in mice, including a large fraction of HSCs that, upon transplantation, replenish blood in a platelet, erythroid and myeloid (PEM)-restricted or strongly biased manner[3,15,16]. Although PEM-restricted or biased HSCs increase with age, they are already prevalent in young adult mice[15,16].

The natural accumulation of nuclear DNA mutations in HSCs of healthy human individuals over time, some being highly recurrent driver mutations providing a clonal advantage without apparent detrimental effects on the development of blood cell lineages, a phenomenon called clonal hematopoiesis (CH)[17,18], offers an opportunity to trace lineage contributions of individual HSC clones in steady-state human hematopoiesis in aged individuals when such HSC clones have expanded to a level at which the lineage contribution of the resultant clonal HSC family, originating from a single ancestral HSC, becomes feasible to assess. While clonal analysis has been performed on blood samples from individuals with nuclear CH mutations[10–12,14], this has not included the critical short-lived erythroid and platelet lineages and has only been investigated at a single time point. If heterogeneous lineage contribution patterns of individual HSC clones are observed at a single time point, this could either reflect individual HSC clones fluctuating in their contribution to different blood cell lineages over time or the existence of HSCs with distinct and stable lineage-biased replenishment patterns. Distinguishing between these possibilities would only be possible if one could reliably track the contribution of the same individual HSC clones to different blood cell lineages over time. While it is not possible to use CH or other somatic nuclear mutations to prospectively trace the lineage contribution of small HSC clones in the bone marrow (BM) of young adults, retrospective inferences from phylogenetic studies of expanded HSC clones in aged individuals[5] are possible, as clonal somatic nuclear mutations defining the expanded clones have been shown to typically be targeted to HSCs several decades before expanding to a traceable clonal size in aged individuals[6,19].

## Results

### A limited repertoire of HSC lineage replenishment patterns

To assess the lineage contribution of HSCs, we investigated the BM of 93 healthy donors (HDs; 24–91 years old, most individuals ≥70 years; for experimental design, see Extended Data Fig. 1 and Supplementary Table 1) with blood parameters within the normal reference range and without any evidence of previous or current hematological disease (Extended Data Fig. 2a–c).

To identify mutations for clonal tracing, we screened DNA from bulk BM mononuclear cells (MNCs) for driver mutations in 23 of the most commonly mutated genes in CH (Methods) by highly sensitive error-corrected targeted DNA capture sequencing (ECTS)[20]. DNA from buccal swabs from 23 donors served as germline controls for removal of SNPs and sequencing errors. A total of 211 somatic CH driver mutations were identified in 71 of 93 individuals (76.3%), with a mean mutant cell fraction (MCF) of 3.03% (range of 0.25–85.2%), with the most frequent mutations observed in DNMT3A followed by TET2, ASXL1 and PPM1D (Extended Data Fig. 2d–g and Supplementary Table 2), in line with previous reports[17,18,21–23]. While detected in 70 of 79 individuals older than 60 years (mean MCF of 3.05%), only in one of 14 individuals 50 years or younger was a driver mutation identified (MCF of 0.80%), also in alignment with previous reports[17,18,21–23].

We screened for additional clonal mutations, including synonymous mutations, mutations in intronic regions and those outside the targeted 23 genes, captured by chance through ECTS. A total of 21 such mutations (mean MCF, 3.52%; range, 0.31–27.1%) of undetermined significance (CH-US) were detected in 19 donors (Extended Data Fig. 2h–l).

The cellular origin and clonal involvement of the hematopoietic stem and progenitor cell (HSPC) hierarchy in CH have yet to be comprehensively investigated. We therefore purified HSCs and distinct progenitors and mature cells of the different blood cell lineages (Extended Data Fig. 3) to assess their involvement in clones marked by a CH driver mutation with an MCF ≥ 2% (51 clones) or by a CH-US mutation with an MCF ≥ 1% (ten clones) in BM MNCs. These 61 clonal mutations were distributed among 33 elderly HDs (median age of 76 years; 70–84 years). Three clones showed no clonal marking of HSCs and only involvement in the T cell lineage, compatible with long-lived T cell clones[24] rather than ongoing hematopoiesis (Extended Data Fig. 4). Of the remaining 58 clonal mutations, all but one were traced back to the Lineage−CD34+CD38−CD90+CD45RA− HSC compartment[25,26] (Fig. 1a and Extended Data Fig. 4), allowing us to use these mutations for clonal lineage tracing by long-term HSC-derived clonal families. Seventeen of the HSC-derived clones (from ten different donors) also underwent phylogenetic analysis after whole-genome sequencing (WGS) of single HSPC-derived colonies, demonstrating that individual clones were established at least 15–71 years ago (mean of 46 years; Fig. 1b), in line with previous studies[5,6,13,19,27], and therefore initiated in long-term self-renewing HSCs.

While several studies have investigated clonal contribution to the myeloid, B cell and T cell lineages, representing the major white blood cell lineages[10–12,14], no previous studies have analyzed the simultaneous contribution of human single-HSC-derived clones to all major blood lineages, also including the critical platelets and erythroid lineages. We therefore examined the contributions of each identified HSC-derived clone (defined by the 47 driver and ten additional mutations identified in HSCs from 31 aged donors; Fig. 1a) to the different blood cell lineages through mutational assessment of purified HSCs, along with mature myeloid, T and B cells as well as erythroid-restricted progenitors (EPs) and megakaryocyte (platelet)-restricted progenitors (MkPs)[25,26], with a median of 2,000 cells investigated for each mature and progenitor cell type, except for MkPs (median of 342 cells) and HSCs (median of 1,478 cells) (Extended Data Fig. 5a).

If multipotent human HSCs can adopt a fate to replenish blood in a lineage-restricted or lineage-biased manner as demonstrated in mice[7,8,15,16], a wide variety of restriction or bias patterns could theoretically have been observed when simultaneously assessing all five major blood cell lineages (platelet, erythroid, myeloid, B and T (PEMBT) cells). However, we only repeatedly observed two patterns of lineage-restricted replenishment among 57 fate-mapped HSC-derived clones. In addition to those contributing to all five lineages (n = 22), we identified single-HSC-derived clones contributing to all lineages except T cells (platelet, erythroid, myeloid and B (PEMB) cells; n = 30) and all three myeloid cells but not B cells and T cells (PEM; n = 5) (Fig. 2a). Among these 57 HSC clones, in addition to those displaying PEM- or PEMB-restricted lineage patterns, there were HSC clones showing strong lineage biases (contribution to one or more lineages more than five times higher than to all other lineages; see Methods for further details) (Fig. 2b). Notably, the only lineage biases observed repeatedly were the same as those observed for lineage-restricted HSCs, namely PEMB- and PEM-biased HSCs. Collectively, PEMB-restricted or -biased (n = 25) or PEM-restricted or -biased (n = 13) HSC lineage contribution was observed more frequently than HSCs contributing to all (PEMBT) lineages in a balanced manner (n = 15) (Fig. 2b). Of the 57 investigated HSC clones, only four showed a lineage replenishment pattern different from that of PEMBT, PEMB or PEM, and each of these was only observed once (Extended Data Fig. 5b).

The different patterns of HSC clonal lineage restriction and bias were independent of clone size in HSCs (Fig. 2c). HSC clones displaying the same lineage restriction or bias pattern differed in contribution to mature cells, and high-productive and low-productive clones for the PEMB lineages were observed regardless of lineage restriction or bias patterns (Fig. 2d). Therefore, mutational lineage tracing of individual HSC clones reveals a limited repertoire of distinct lineage restriction and bias patterns in aged human hematopoiesis.

While the impact of CH driver mutations on clonal expansion is well established[19,27,28] and, in some instances, their effect on mature blood cells has been implicated[21,29–31], there are no findings to suggest that CH

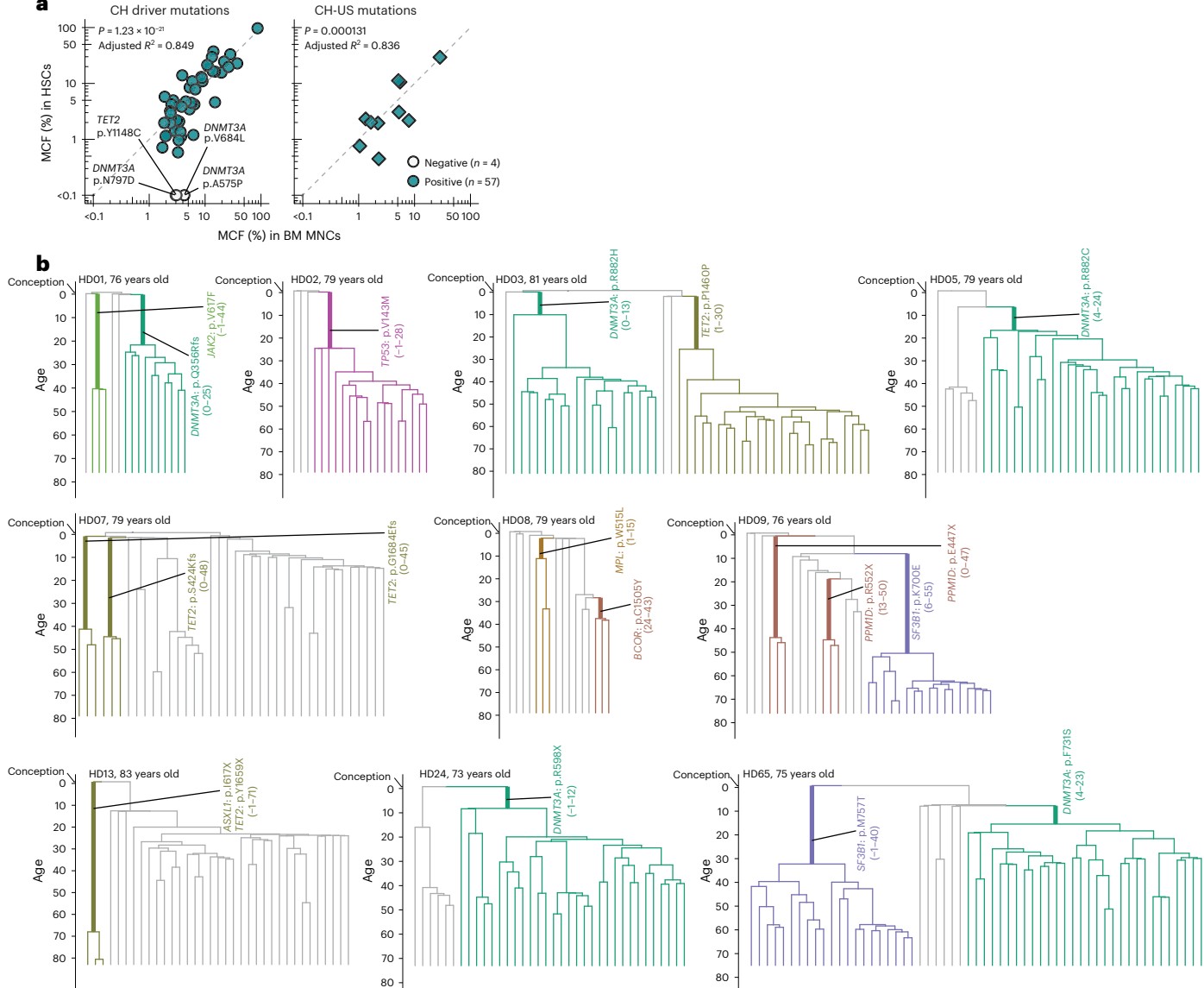

**Fig. 1 | Dating the origin of long-term HSC clones. a**, Mutations with BM MNC MCF ≥ 2% for CH driver mutations and ≥1% for other CH-US mutations were used for clonal lineage tracing. Panel shows MCF of the CH driver (left, $n = 51$) and CH-US (right, $n = 10$) clonal somatic mutations in Lineage⁻CD34⁺CD38⁻CD90⁺CD45RA⁻ HSCs compared to BM MNCs. Mutations not confidently detected in HSCs ($n = 4$) are shown with open symbols. $R^2$ values adjusted for the number of predictors for a linear model fit and $P$ values for Pearson correlation analysis (two-sided test) adjusted with Benjamini–Hochberg method correction for multiple comparisons are shown. **b**, Phylogenetic trees mapping 17 clonal mutations targeted to HSCs in ten HDs (HDs 01, 02, 03, 05, 07, 08, 09, 13, 24 and 65) constructed from WGS results of in vitro expanded clones

from single HSPCs without (gray) and with (colored) CH mutations identified through ECTS. Branches that share the same CH mutation are highlighted in bold. The position and length of each branch represent the estimated time period when the mutation occurred. The $y$ axis in the trees is scaled based on donor age, starting (top of the phylogenetic tree) at the time of conception (40 weeks before birth) and ending (bottom of the tree) at the time of collection of the investigated BM sample. The 95% confidence intervals of the age (years) of branches are shown in parentheses next to the CH mutation names. WGS analysis confirmed that the clone marked by the synonymous CH-US mutation *TET2* p.P1460P in HD03 was not accompanied by other known CH driver mutations.

driver mutations result in complete loss of HSC replenishment of any blood cell lineages[32–35]. In line with this, individuals with CH, including those with very large clones, sustain normal blood parameters[17,22,36] (Extended Data Fig. 2b). Among the 25 PEMB-restricted or -biased and 13 PEM-restricted or -biased HSC clones, different recurrent CH driver as well as CH-US mutations were widely distributed (Fig. 2e) and with a prevalence corresponding to the screening for CH driver mutations in our study (Extended Data Fig. 2d,e) and previous studies[17,18,21–23]. In line with this, there was no significant bias for any specific CH driver mutation when comparing PEMB- and PEM-restricted and -biased HSC clones (Fig. 2e). Combined with both PEMB and PEM HSC clones including clones marked by CH-US mutations, this strongly supports

that PEMB- and PEM lineage restriction and bias are not primarily determined by CH driver mutations.

In contrast to what we observed for PEMB and PEM HSC clones, HSC clones with a balanced contribution to PEMBT cell lineages were highly enriched in clones originating from HSCs targeted by *DNMT3A* mutations (Fig. 2e). This is in agreement with previous reports showing higher T cell involvement in *DNMT3A*-initiated clones[10,12], compatible with promoting T cell replenishment from aged HSCs and/or increasing the lifespan or expansion of mature T cells. An alternative explanation, in line with our phylogenetic analysis showing that it takes decades from when an HSC is targeted by a CH driver mutation until its resultant HSC clonal family has expanded sufficiently to become detectable and reliably

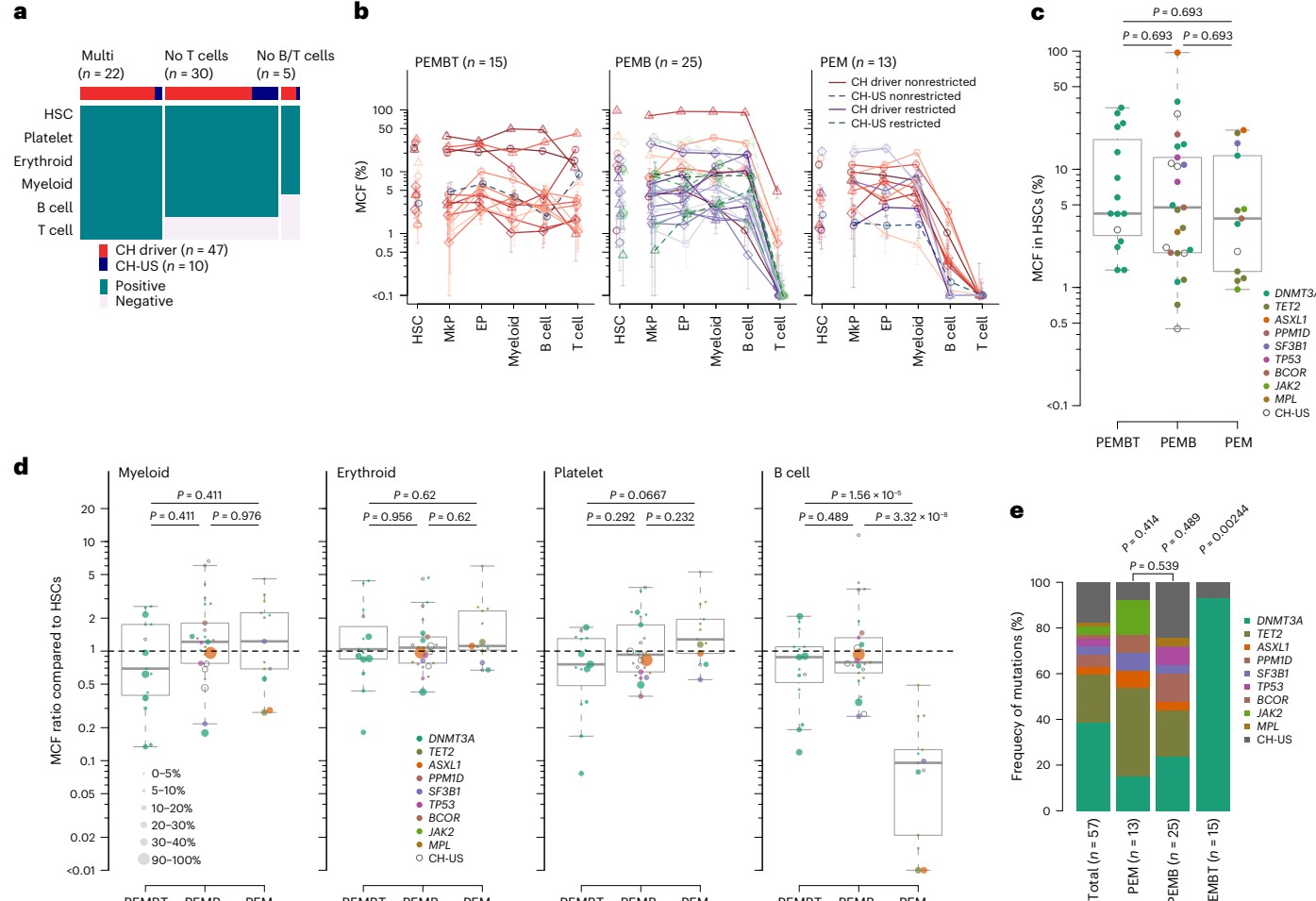

**Fig. 2 | Clonal fate mapping identifies HSCs with restricted and biased contribution to different blood cell lineages. a**, Presence or absence of mutations originating in HSCs across platelet, erythroid, myeloid, B and T cell lineages, presented as those involving all lineages (Multi), all lineages except T cells and all lineages except B and T cells. HSCs (Lineage⁻CD34⁺CD38⁻CD90⁺CD45RA⁻) and MkPs (Lineage⁻CD34⁺CD38⁺CD41⁺) for the platelet lineage; EPs (Lineage⁻CD34⁺CD38⁺CD41⁻CD123⁻CD45RA⁻) for the erythroid lineage; myeloid cells (CD33⁺CD14⁺), B cells (CD19⁺) and T cells (CD4 or CD8 single positive). Positivity was judged using the overlapped area of posterior distributions by Bayesian inference using the number of sorted cells and the number of events by droplet digital polymerase chain reaction (ddPCR) between the targeted sample and the normal control (see Methods for details). No other lineage replenishment patterns than those shown were reproducibly observed. **b**, Multipotent balanced (PEMBT), lineage-biased (mapped by CH driver mutations, solid lines with red or orange; other CH-US mutations, dashed blue colors) and lineage-restricted (driver, solid purple; CH-US, dashed green) contribution patterns observed from single-HSC-derived clones based on percent contribution (percent mean MCF ± 95% credible interval estimated with the Bayes method using the number of ddPCR events and the number of cells analyzed; Methods) to the PEMBT cell lineages. Lineage bias patterns were categorized by comparing the ratio of MCF in each lineage (see Methods for definition of bias patterns). Only HSC lineage bias patterns observed for more than one HSC clone are shown. Four HSC clones with unique patterns (each observed only once) are shown in Extended Data Fig. 4. **c**, HSC MCF (%) for somatic CH driver and CH-US mutations associated with

PEMBT (*n* = 15), PEMB (*n* = 25) and PEM (*n* = 13) lineage replenishment patterns. Lineage-biased and -restricted PEMB and PEM clones were pooled. Center lines and boxes denote median values and the first and third quartiles, respectively. Whiskers indicate maximum and minimum values within 1.5× interquartile range. Mutated driver genes are indicated with different colors. Statistical tests were performed using two-sided Wilcoxon rank-sum test, and *P* values were adjusted with the Benjamini–Hochberg method. **d**, MCF ratio for all mutations shown in **c** for the indicated lineages compared to HSCs for PEMBT (*n* = 15), PEMB (*n* = 25) and PEM (*n* = 13) clones. Lineage-biased and -restricted PEMB and PEM clones were pooled. Individual mutations are indicated with a colored bubble; bubble size represents the MCF in HSCs, and the color indicates the targeted gene, as indicated in the legends. Center lines and boxes denote median values and the first and third quartiles, respectively. Whiskers indicate maximum and minimum values within 1.5× interquartile range, and outliers beyond these are plotted. Statistical tests were performed using two-sided Wilcoxon rank-sum test, and *P* values were adjusted with the Benjamini–Hochberg method separately for each lineage. **e**, Distribution of CH driver and CH-US mutations in PEM, PEMB and PEMBT HSC clones. Lineage-biased and -restricted PEMB and PEM clones were pooled. The distribution of mutated genes in each lineage pattern was compared against that of the overall distribution of mutations for the frequency of (1) *DNMT3A*, (2) *TET2* and (3) other mutations (combined) by two-sided Fisher's exact test. Mutational distribution within PEM and PEMB patterns was also compared. *P* values were adjusted with the Benjamini–Hochberg method.

fate mapped (Fig. 1b), would be that the prevalent *DNMT3A*-mutated PEMBT HSC clones have emerged earlier in life than the PEMB- and PEM-restricted and -biased clones. This would also be compatible with T cell replenishment from HSCs declining with age[37] and T cells having a much longer lifespan than short-lived myeloid blood cell lineages[24,38]. In support of this, previous studies implicated a weaker fitness advantage

of *DNMT3A* mutations than that of other CH driver mutations[19,27,28]. Thus, we performed phylogenetic analysis of *DNMT3A*-mutated PEMBT clones and confirmed that these had emerged significantly earlier in life than PEMB and PEM clones with non-*DNMT3A* CH driver mutations (Fig. 3a). If *DNMT3A* mutations do not significantly affect HSC replenishment of T cells, we further argued that balanced PEMBT HSC

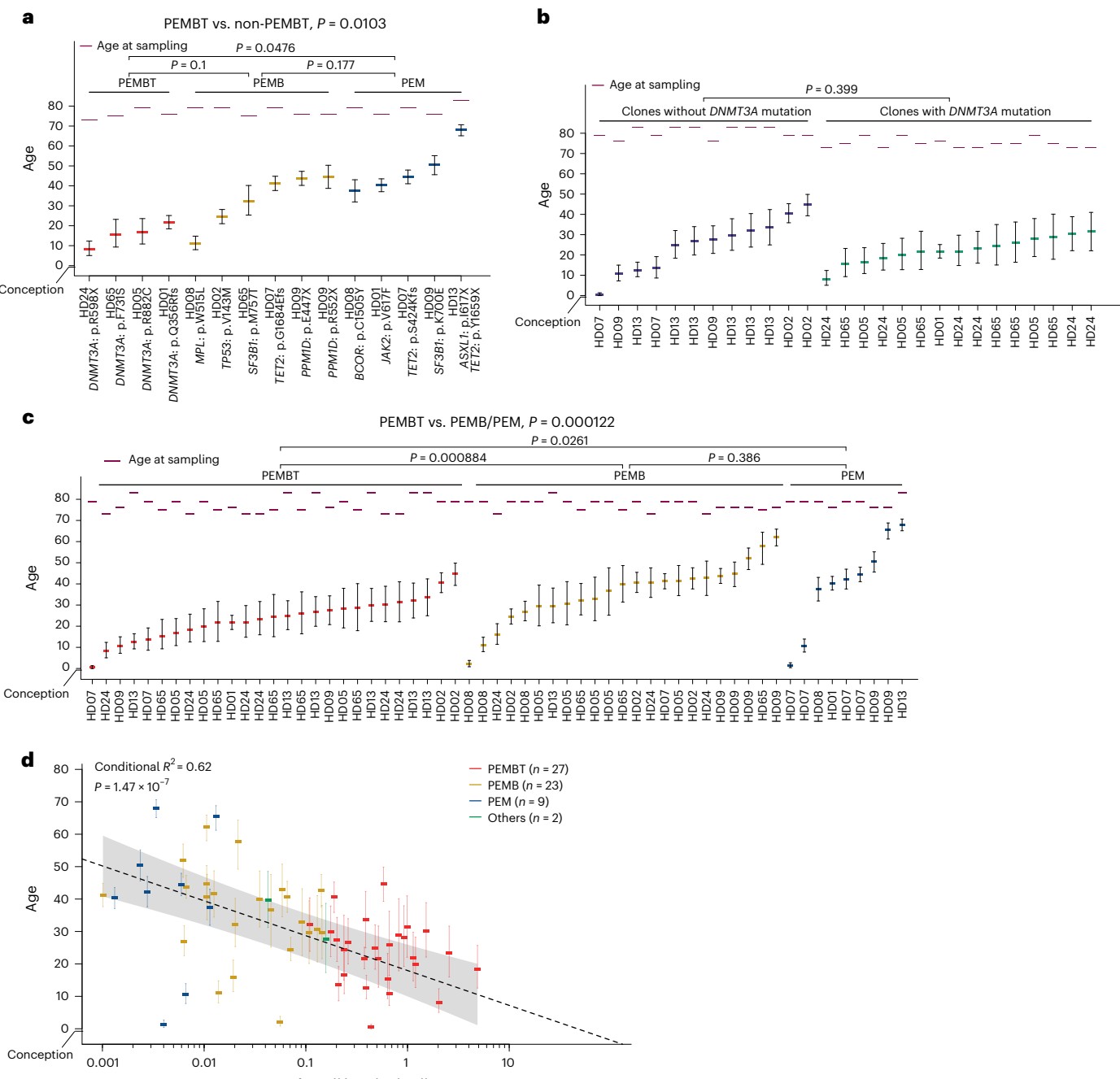

**Fig. 3 | Dating of mutational acquisition in HSC clones with different lineage replenishment patterns. a**, Dating of acquisition of CH mutations defining HSC clones categorized as PEMBT ($n = 4$), PEMB ($n = 6$) or PEM ($n = 5$) at the time of BM analysis. Lineage-biased and -restricted PEMB and PEM clones were pooled. The $y$ axis denotes the number of years after conception when the specified mutations were acquired by a single HSC followed by clonal expansion. Colored lines represent the estimated timing of nodes of branches when CH mutations were acquired, extracted from the phylogenetic trees (Fig. 1b), and error bars represent 95% confidence intervals, estimated from the number of mutations on each branch (see Methods for details). Purple lines denote the age at sampling. Ages at the nodes were compared between PEMBT, PEMB and PEM patterns using pairwise two-sided Wilcoxon rank-sum tests, with $P$ values adjusted using the Benjamini–Hochberg method. The comparison between PEMBT and non-PEMBT patterns was conducted using a Wilcoxon rank-sum test. **b**, Clones with PEMBT patterns shown in Supplementary Figs. 1–3 are displayed in the same manner as in **a**, according to the presence ($n = 15$) or absence ($n = 12$) of

*DNMT3A* mutations. Subclones that branched after the acquisition of a *DNMT3A* mutation were treated as *DNMT3A*-mutant clones (Supplementary Fig. 1). The comparison between PEMBT clones with and without a *DNMT3A* driver mutation was conducted using a two-sided Wilcoxon rank-sum test. **c**, All clones shown in Supplementary Figs. 1–3 are displayed in the same manner and were subjected to the same statistical tests as in **a**, according to lineage contribution patterns (PEMBT, $n = 27$; PEMB, $n = 23$; PEM, $n = 9$). **d**, MCF ratio for T cells compared to myeloid cells and branch acquisition timing for HSC clones with PEMBT, PEMB and PEM patterns in **c** and Supplementary Figs. 1–3. Lineage-biased and -restricted PEMB and PEM clones were pooled. The dashed line and the gray shaded area represent the regression line and its 95% confidence interval, respectively, of the ages of the branch nodes, which are indicated with dense color bars, estimated using a linear mixed-effect model. Conditional $R^2$ value (adjusted for the number of predictors for model fit) and $P$ value from a two-sided $t$-test for the regression coefficient are provided.

clones lacking *DNMT3A* mutations should also have emerged earlier in life. We assessed the lineage contribution of major clades identified by single-cell-derived colonies in ten donors subjected to WGS (Extended Data Figs. 6–8) and identified multiple *DNMT3A*–wild-type (as well as additional *DNMT3A*-mutated) balanced PEMBT clones. The timing of clonal mutation acquisition overlapped between PEMBT clones with and without *DNMT3A* mutations (no statistical difference; Fig. 3b). Collectively, our findings support that *DNMT3A*-mutated HSC clones in older individuals frequently showing balanced contribution to all major blood cell lineages, including T cells, reflect that the clonal mutation was acquired early in life when most HSCs actively replenish long-lived T cells that subsequently can persist for life.

While our WGS-based clonal and phylogenetic analysis of HSC clones in aged individuals demonstrated that multipotent PEMBT HSCs dominate early in life and become increasingly replaced by PEMB- and PEM-restricted and -biased HSCs upon aging, this analysis also established that HSC clones can continue to actively contribute to replenishment of all major blood cell lineages (including T cells) for decades (Fig. 3a,c–d). Likewise, whereas PEMB- and PEM-restricted and -biased HSC clones (with no or little evidence of having produced long-lived T cells) increase and expand with age, they can emerge already early in life (Fig. 3a,c–d).

### HSC clonal lineage restriction is intrinsically programmed

While the myeloid (PEM) lineages represent short-lived blood cells in need of continuous replenishment, B lymphocytes are in part long lived[39]. Therefore, the observed B cell contribution by most HSC clones in aged healthy individuals could reflect the persistence of long-lived B cells rather than ongoing B lymphopoiesis, as observed for T cells. However, our phylogenetic analysis showed that PEMB clones emerge later than PEMBT clones, overlapping with the emergence of PEM clones (Fig. 3a,c), compatible with ongoing B cell replenishment at a later age. To establish to what degree HSC clonal contribution to B cells in aged healthy individuals reflect ongoing B lymphopoiesis or persistence of long-lived B lymphocytes, we investigated BM of aged HDs ($n = 11$, 74–87 years old) for clonally derived CD34+CD19+ B cell progenitors (pro-B cells; Extended Data Fig. 3a). From four PEM-restricted HSC clones with no mature B cells, there was also no contribution to pro-B cells, whereas, for 14 of 15 investigated HSC clones with mature B cell contribution, CD34+CD19+ pro-B cells were also clonally involved (Fig. 4a) at similar MCFs as mature B cells (Fig. 4b), demonstrating that B cell contribution from individual HSC clones in aged individuals largely reflects persisting B lymphopoiesis.

To investigate whether the inability of PEM-restricted HSCs to replenish B cells at steady state reflects an intrinsically programmed rather than extrinsically regulated property, we transplanted CD34+ or CD34+CD19− BM cells into NOD.Cg-*Prkdc*^SCID^;*Il2rg*^tm1Wjl^/SzJ (NSG) mice highly permissive for reconstitution of human B lymphopoiesis[40]. We selected three aged HDs from whom we could simultaneously mutationally fate map coexisting PEMB-restricted and PEM-restricted HSC clones. The BM of transplanted mice was analyzed after 10–14 weeks for clonal contribution to myeloid and B cells (Extended Data Fig. 9). For four HSC clones contributing robustly to pro-B cells in steady-state hematopoiesis in the HDs, a clonal B cell contribution was observed upon transplantation, corresponding to levels seen in the BM of the donor (Fig. 4c–e). By contrast, for all four individual HSC clones showing no evidence of steady-state B cell contribution in the donor BM, there was also no detectable contribution to B cells in the mice (Fig. 4c–e), although other HSPCs from the same transplanted BM cells effectively replenished human CD19+ B cells in the mice (Extended Data Fig. 9), and the traced clones did contribute to myelopoiesis as in the healthy participants (Fig. 4c–e). These findings confirm that aged HSCs contribute to active B lymphopoiesis and that the steady-state PEM restriction or bias of a subset of human HSCs is intrinsically programmed.

### Phylogenetic analysis infers stable HSC lineage replenishment

WGS of single-cell-derived colonies enables reconstruction of phylogenetic trees to establish the timing and lineage relationships between ancestral clones and their descendant subclones[5,6]. Through such phylogenetic analysis, we established that all investigated clones with CH driver mutations emerged from a single HSC decades ahead of the BM analysis (Fig. 1b). This included two PEM-restricted clones defined by a *TET2* p.S424Kfs mutation and an *ASXL1* p.I617X and *TET2* p.Y1659X CH driver mutation from HD07 and HD13, respectively (Fig. 5a and Extended Data Fig. 7). That these mutations had been targeted to a single HSC decades ago suggests that the observed PEM-restricted fate had already been determined at the time when the clonal mutation was acquired and propagated to all subsequent HSC offspring and subclones established through decades of HSC expansion, clonal evolution and diversification, without at any stage having produced long-lived T cells. To obtain further experimental evidence for this and for PEM lineage restriction not being caused solely by CH driver mutations, we evaluated lineage contributions of clones identified through WGS that lacked known driver mutations (CH-US) and were established several decades ago. This revealed two additional PEM-restricted clones in HD09 and HD07 (Fig. 5a and Extended Data Fig. 7), of which the latter was a sibling of the above *TET2* p.S424Kfs PEM-restricted clone, both having emerged from a common PEM-biased clone (Fig. 5b), further supporting that the observed PEM lineage fate restriction was programmed decades ago and unlikely to be primarily the result of specific driver mutations.

In further support of *DNMT3A* mutations having little or no impact on T cell replenishment from HSCs, but rather T lineage contribution of HSCs diminishing with age (Fig. 3), phylogenetic analysis of *DNMT3A*-mutated (Fig. 5c) as well as *DNMT3A*-non-mutated (Fig. 5d) PEMBT clones revealed subclones emerging decades later with reduced or absent T cell contribution compared to the parental clone. Reduction in T cell contribution was also observed in descendant subclones of ancestral PEMB-biased clones (Extended Data Fig. 10).

We next evaluated changes in lineage contribution over time across all ancestor clonal and descendant subclonal pairs within phylogenetic clades (Extended Data Figs. 6–8 and summarized in Fig. 5e–j). Two relationships were observed between ancestral clones and descendant subclones. The first 'hierarchical' pattern revealed descendant subclones with a higher degree of lineage restriction or bias (from PEMBT toward PEMB or PEMB toward PEM) than the ancestral clone (Fig. 5e–f). The second and most common 'stable' pattern revealed descendant clones that, despite decades of separation from their ancestral clone, showed highly concordant lineage contribution patterns (Fig. 5g–h) ($P = 0.0026$ by one-sided permutation test assessing concordance among siblings; one-sided Moran's *I*-test assessing correlation with phylogenetic distance, see Supplementary Table 3 for *P* values and statistics). In addition, while the subclonal dissection of HSC families potentially could have uncovered descendant HSC subclones with distinct lineage replenishment patterns, only the same (PEMBT, PEMB and PEM) patterns were reproducibly observed, as for the ancestor clones.

### Prospective analysis reveals stable HSC lineage replenishment

Phylogenetic lineage analysis inferred that HSC subclones within a clonal HSC family frequently are highly stable with regard to lineage replenishment pattern, despite decades of expansion and clonal evolution (summarized in Fig. 5e–j). However, such retrospective inferences are limited by being based on lineage analysis of a single BM sample, and, to date, the steady-state contribution of human as well as mouse HSC clones to different blood cell lineages has only been investigated at single time points[7,8,10–14], precluding assessment of whether distinct lineage restrictions or biases reflect a fluctuating or a stable HSC property. Addressing these fundamental questions about functional properties of HSCs in steady-state hematopoiesis requires prospective lineage

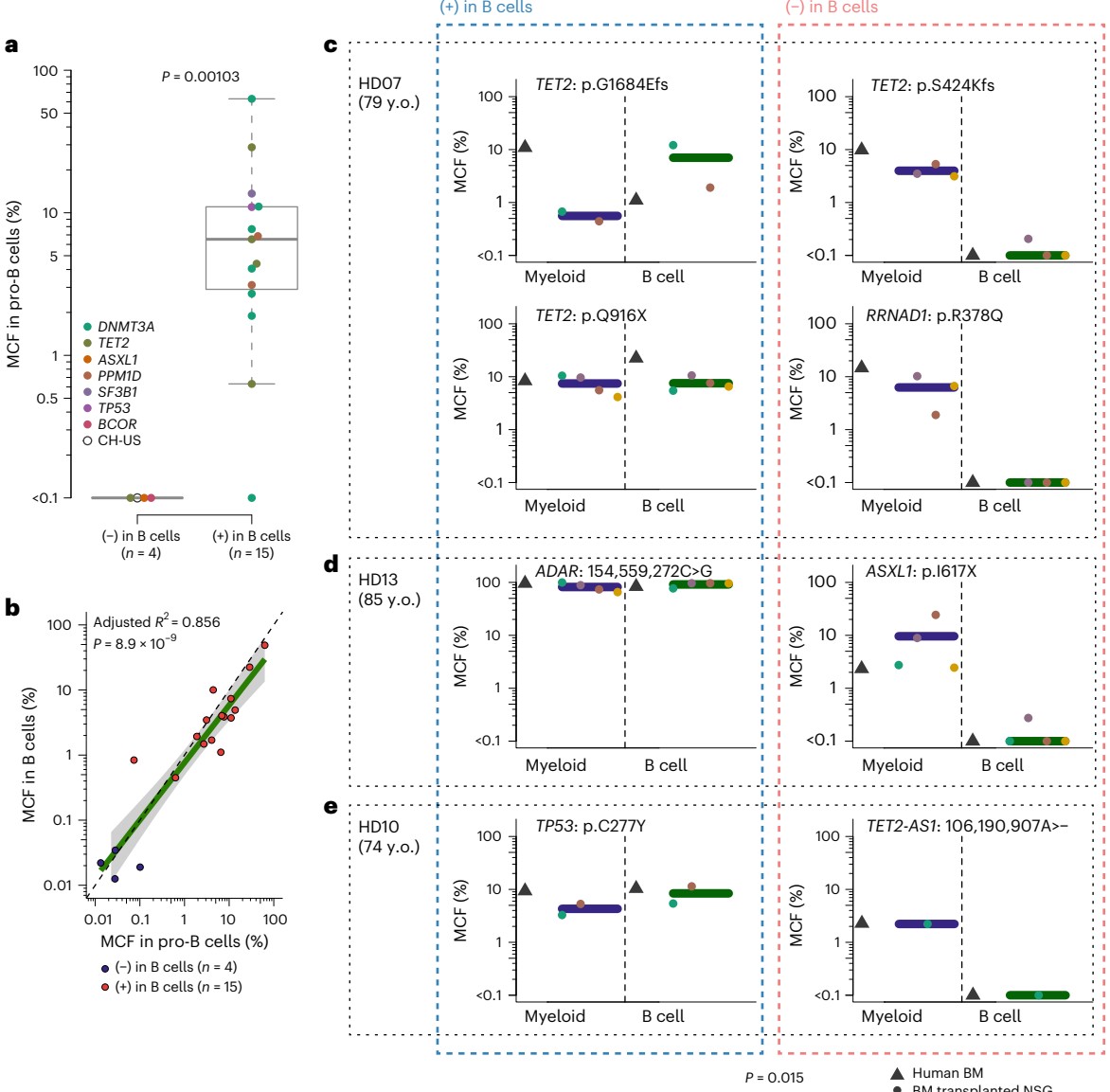

**Fig. 4 | Clonal HSC contribution to B cells reflects ongoing B lymphopoiesis.** **a**, MCF in BM Lineage⁻CD34⁺CD19⁺ pro-B cells of individual HSC-derived clones without (*n* = 4) and with (*n* = 15) mature B cell contribution. Center lines and boxes denote median values and the first and third quartiles, respectively. Whiskers indicate 1.5× interquartile ranges, and outliers beyond these bounds are plotted. *P* value was evaluated with the two-sided Wilcoxon rank-sum test. **b**, Correlation between contribution to mature B cells and BM Lineage⁻CD34⁺ CD19⁺ pro-B cells of individual HSC-derived clones without (*n* = 4) and with (*n* = 15) mature B cell contribution. The green line and gray shading represent the regression line and its 95% confidence interval, respectively. $R^2$ value (adjusted for the number of predictors for the linear model fit) and *P* value for Pearson correlation analysis (two-sided test) are provided. **c–e**, Contribution

(percent MCF) of individual HSC-derived clones to myeloid and B cell lineages in the BM of HDs 07, 13 and 10 (gray triangles) and corresponding data from the BM of NSG mice (each circle represents a separate mouse) 10–14 weeks after transplantation of the corresponding donor CD34⁺ (HD07 and HD13) or CD34⁺CD19⁻ (HD10) cells. Only mice in which the tracked clone (mapped by the indicated mutation) was confidently detected in myeloid cells are shown. Multiple B cell samples (2,000 cells per sample, *n* = 2–3) were isolated from each NSG BM sample and used as technical replicates for each mouse, and the average values are plotted. y.o., years old. Horizontal bar indicates the mean for all mice. *P* value provided at the bottom was assessed with a binomial mixed-effect model (two sided) to evaluate the association of B cell and myeloid cell positivity before and after transplantation (see Methods for details).

fate mapping of the same HSC clonal families over time. We therefore serially collected (at two to four time points) BM from 11 aged HDs for up to 65 months, enabling prospective tracking of PEMBT cell lineage contribution of 22 HSC clonal families (Fig. 6). For all 22 HSC clones, the pattern of lineage replenishment was remarkably stable over time (Fig. 6a–e), whether representing balanced PEMBT, PEMB-restricted or -biased or PEM-restricted or -biased HSCs (Fig. 6a–c) or unique clonal HSC lineage bias patterns (Fig. 6d). This clonal stability was observed regardless of different CH driver or CH-US mutations and included clones marked by CH-US mutations identified through single-colony WGS (Fig. 7). The stability of HSC clones was not restricted to lineage

replenishment patterns, but was also observed with regard to clonal size (Fig. 6a–d), which might in part reflect that the yearly clonal expansion induced by most CH mutations is very limited[19,27,28] and that, in aged individuals with oligoclonal hematopoiesis, CH clones are likely to compete with other clones that also possess a clonal advantage.

## Discussion

The high number (50,000–200,000) of human HSCs, each contributing minimally to hematopoiesis[5,6], prohibits reliable assessment of steady-state contribution of individual HSC clones in young adults to the multiple critical blood cell lineages they possess the potential to

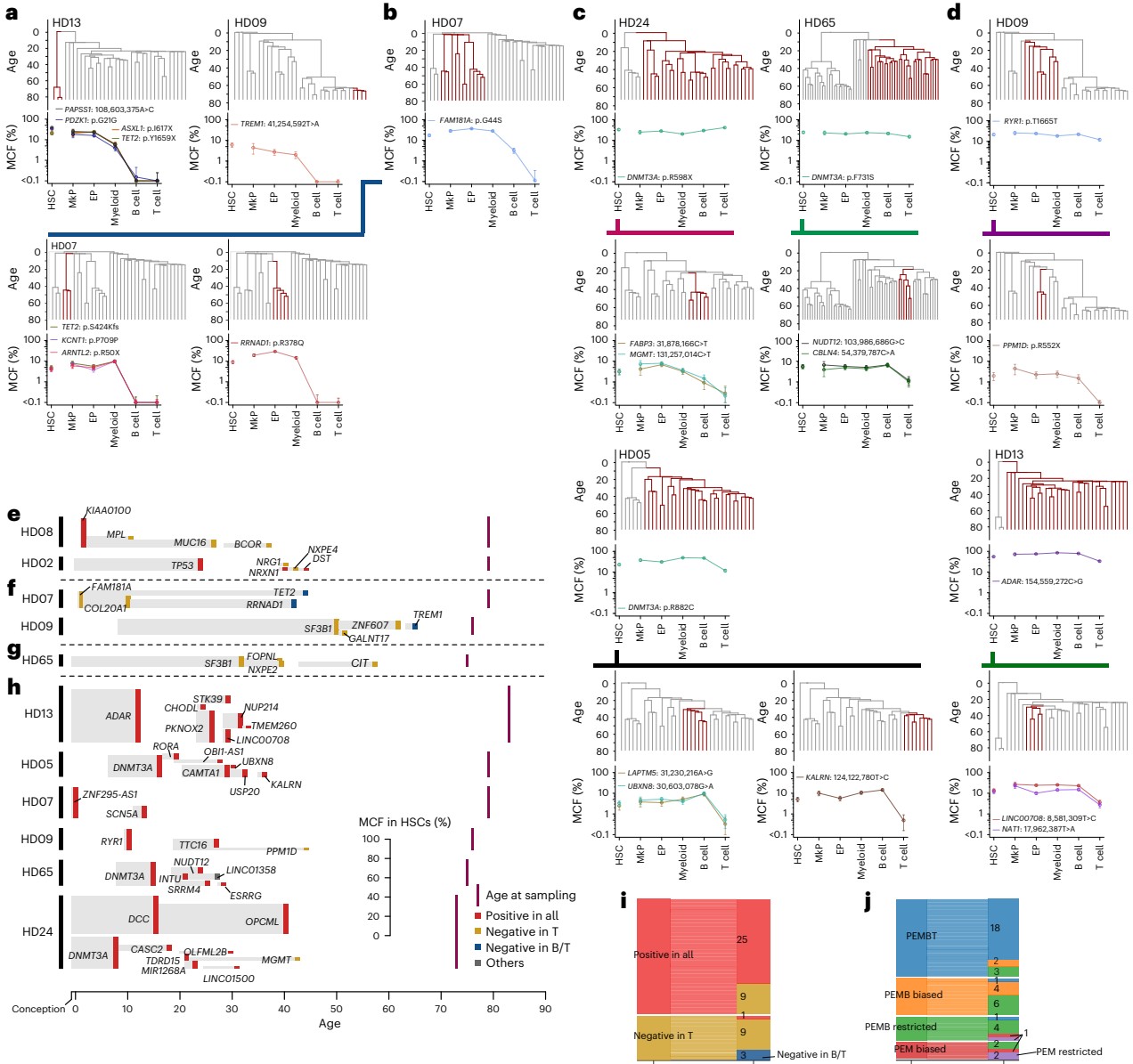

**Fig. 5 | Phylogenetic histories of PEM-restricted HSC clones implicate programming of lineage restriction decades ago. a**, Mean MCFs in the different blood cell lineages for four HSC clones with a PEM-restricted lineage contribution pattern from three donors (HD13, HD09 and HD07) (error bars show 95% credible interval estimated with the Bayes method taking into account the number of droplets and the number of input cells (see Methods for details)). Clades marked by each specified mutation on the phylogenetic tree are highlighted in red in the trees above each clonal plot. *RRNAD1* (official gene name: *METTL25B*). The *y* axis in the trees is scaled based on donor age, starting (top of the phylogenetic tree) at the time of conception (40 weeks before birth) and ending (bottom of the tree) at the time of collection of the investigated BM sample, as in Fig. 1b. **b**, MCFs in the different blood cell lineages and 95% credible intervals estimated for the common ancestor clone of two PEM-restricted clones shown in **a** from donor HD07. **c,d**, MCFs in the different blood cell lineages and 95% credible intervals for *DNMT3A* mutation-positive (**c**) and -negative (**d**) ancestral HSC clones contributing to all five lineages including T cells and their descendant clones showing reduced or absent contribution to T cells. Ancestors and their descendant clones are connected by lines of different colors for each donor. **e–h**, Changes in lineage contribution over time by tracking ancestor and descendant HSC clones identified by single-colony WGS. **e**, HSC clones originally contributing to all five blood lineages that gave rise to new subclones no longer contributing to the current T cell pool. *KIAO100* (Official gene name: *BLTP2*). **f**, PEMB HSC clones originally contributing to all lineages except T cells, which gave rise to new subclones no longer contributing to the T cell or B cell

pool. **g**, HSC clone established three decades ago in which the ancestral clone as well as the descendant subclones established one to three decades later stably contributed to all PEMB lineages but not the current T cell pool *FOPNL* (Official gene name: *CEP20*). **h**, Multilineage (PEMBT) HSC clones established decades ago for which multiple subclones emerging decades later continued to contribute to all lineages, including T cells. For all subpanels, height of clones corresponds to the size (MCF) of the ancestral HSC clone at the time of sampling, and the heights of bars to the right indicate the size of subclones marked by the indicated subclonal HSC mutation. The *x* axis indicates the age of the donor and the timing of each of the clonal mutations. Lineage contribution of ancestral and descendant HSC clones is indicated by the colored bars at the nodes of the branches where the mutations occurred. Gene names analyzed by ddPCR and used in Fig. 3c are labeled on the corresponding branches. The age of the donor at the time of BM sample collection is indicated by the position of the purple bars at the right of the subpanels. *LINCO1358* (official gene name: *JUN-DT*). **i**, Alluvial plot showing positivity in mature cell lineage contribution for paired ancestor and descendant clones (*n* = 47), based on phylogenetic trees derived from WGS analysis of single-cell colonies (summarized from **a**–**d** and Supplementary Figs. 1–3). Ancestors with multiple descendant subclones are presented once for each subclone. Clones shown are those that contributed to all five lineages, negative in T cells or negative in B and T cells. **j**, Alluvial plot that (unlike **i**) takes into account changes in lineage biases (Methods) as well as lineage restrictions, comparing PEMBT, PEMB and PEM lineage contribution patterns of paired ancestor and descendant HSC clones (*n* = 46).

replenish. A study in young adult individuals used mitochondrial mutations to trace lineage contribution of clonally related HSC groups[41]. These HSC groups were considerably larger than HSC clones identified through clonal nuclear mutations to have emerged in young adults[5,6], compatible with the tracing of larger HSC clones having emerged already during fetal development rather than in adult BM[6]. Moreover, all investigated 'clonal groups' showed contribution to all investigated blood cell lineages and with only minor lineage bias[41], in further agreement with representing clones derived from fetal HSCs, as single-mouse HSC transplantation studies have established that lineage-restricted HSC clones do not emerge until after birth[42]. In addition, this previous study investigated total lymphocytes[41] rather than B and T lymphocytes separately, therefore failing to reveal the potentially distinct contribution patterns to these two key lymphocyte lineages.

In agreement with previous studies[5,6], the clonal nuclear DNA mutations we used to lineage fate map expanded HSC clones in aged individuals could (through phylogenetic analysis) be backdated to have targeted a single HSC decades earlier, typically in young adult life[6]. Along with very few young adults having detectable CH mutations[17,18], this confirms that reliable clonal lineage fate mapping of adult HSCs is only feasible after a long period of competitive expansion. Thus, by the time that a clone replenished from a single human HSC has become traceable through mutational lineage fate mapping, one is tracing the lineage contribution of an expanded clonal family of HSCs derived from a single HSC rather than the output of a single HSC. In steady-state human hematopoiesis, this is largely restricted to aged individuals in whom extensive and measurable contribution to steady-state hematopoiesis by expanded HSC clones is the rule rather than the exception. This allowed us to address clonal HSC contribution to all five major blood lineages in aged individuals through prospective serial lineage analysis of the same HSC clones over time as well as through retrospective inferences made through phylogenetic analysis.

Assessing clonal HSC contribution to all five major blood cell lineages in aged individuals, we only observed three reproducible patterns. In addition to HSC clones having replenished all five lineages in a balanced manner (PEMBT clones), this included two lineage-restricted or extensively lineage-biased PEMB and PEM patterns. While our studies were limited to studies of aged individuals with CH, the WGS-based clonal and phylogenetic analysis demonstrated that multilineage (PEMBT) replenishing HSCs dominate early in life and become increasingly replaced by PEMB- and PEM-restricted and -biased HSCs upon aging. The extended phylogenetic analysis also established that HSC clones can continue to actively contribute to replenishment of all the major blood cell lineages (including T lymphocytes) for decades. Likewise, whereas PEMB- and PEM-restricted and -biased HSC clones (with no or little evidence of having ever produced long-lived T cells) increase and expand with age, our phylogenetic data suggest that they can emerge already early in life, either by being fate biased from the beginning or alternatively by an ancestral HSC being programmed to become lineage biased, with all HSC daughters subsequently becoming lineage biased in a similar fashion and time frame. These findings coincide with our main finding upon phylogenetic lineage analysis, namely that the lineage replenishment pattern of most HSC clonal families, including their descendant subclones, remains stable over decades, despite extensive clonal HSC expansion and subclonal branching. This remarkable stability of HSC clonal lineage replenishment was confirmed upon transplantation into immune-deficient mice as well as through prospective clonal HSC lineage analysis over years. Collectively, these findings implicate an HSC-intrinsic mechanism propagated effectively to descendant HSCs upon clonal HSC expansion, most likely involving epigenetic cues. While stable HSC clonal lineage replenishment was the rule, we also observed several instances in which descendant subclones, when compared to their ancestral clone, revealed a higher degree of lineage restriction or bias (from PEMBT toward PEMB or PEMB toward PEM pattern), establishing HSC hierarchical relationships.

Importantly, PEMB- and PEM-restricted and -biased HSC clones were randomly distributed among HSC clones targeted by different CH driver and CH-US mutations. Moreover, through phylogenetic analysis, we could show that both ancestor and sibling clones of a PEM-restricted clone with a known CH driver mutation showed the same PEM restriction despite not being targeted by a CH driver mutation. While CH mutations are likely to impact on HSC lineage replenishment, as previously reported[32–35], our findings suggest that they are unlikely to be the primary determinants of the observed distinct clonal HSC lineage replenishment patterns.

Aging is known to be associated with systemic low-grade inflammation[43], which has been shown to influence hematopoietic lineage output[44,45]. Likewise, sex-related differences in lineage composition have been reported[46,47]. Because our study cohort consisted primarily of older individuals with a female preponderance, we could not assess the impact of these factors on the observed lineage replenishment patterns.

Similar findings in normal young adult mice[15] provide supportive evidence that the lineage restriction patterns observed in aged humans are not unique to old individuals and are not caused primarily by CH mutations. The number of HSCs contributing to steady-state hematopoiesis in mice is also high, and the phenomenon of considerably expanded HSC clones during the short lifetime of mice is uncommon[9]. However, in mice, there is a unique experimental setting in which the lineage contribution of a single-HSC-derived clonal HSC family can be assessed with very high specificity by transplanting a traceable, normal single HSC into a myeloablated congenic recipient along with a minimal number of competing normal HSCs[15]. This approach thereby

---

**Fig. 6 | Distinct HSC clones sustain stable balanced or lineage-biased blood replenishment over years. a–d**, Mean MCFs in HSCs and different blood cell lineages (indicated by colored bars) in serially collected BM samples (error bars show 95% credible intervals estimated with the Bayes method taking into account the number of droplets and the number of input cells (see Methods for details)) replenished by clonal PEMBT HSCs (**a**), PEMB HSCs (**b**), PEM HSCs (**c**) or HSCs with unique (each only observed once) lineage patterns (**d**). The mutations tracked and the time (mo, months) between samples are specified above bars. *P* values from two-sided *z*-tests assessing the correlation between clone sizes in serial samples and those in the initial-visit samples, estimated using mixed gamma models, are provided for each pattern (see Methods for details). *P* values were adjusted with the Benjamini–Hochberg method for multiple comparisons. Similarly, tests were conducted stratifying individually for each cell lineage (MkP, $P = 2.2 \times 10^{-4}$; EP, $P = 0.0011$; myeloid cell, $P = 2.6 \times 10^{-6}$; B cell, $P = 2.2 \times 10^{-4}$; T cell, $P = 1.9 \times 10^{-8}$). Gene names and specific amino acid changes (for coding mutations) or positions in the genome and nucleotide changes (for noncoding mutations) are specified for each clone. Of the 11 donors subjected to serial BM analysis, eight (HDs 04, 06, 07, 08, 09, 10, 11 and 12) had more than one traceable HSC clone. HD13 had two mutations (*ASXL1*, p.I617X and *TET2*, p.Y1659X) within the same HSC clone (Fig. 1b). Lineage contribution patterns were assigned to each clone based on the MCFs in the first BM sample. In **b–d**, the restricted clones (#) are presented first followed by biased patterns (§). **e**, Bubble plots showing the ratio of MCF in platelet (top row), erythroid (top middle row), myeloid (bottom middle row) and B (bottom row) blood cell lineages over time relative to the MCF in HSCs for the individual PEMBT, PEMB and PEM clones. Lineage-biased and -restricted PEMB and PEM clones were pooled. Each line represents a separate clone. Circle size and error bar at each analysis time point indicate the MCF (%) in HSCs and credible intervals estimated with the Bayes method, respectively. *P* values from two-sided *z*-tests for the correlation between relative clone sizes in serial samples and those in the initial-visit samples, estimated with mixed gamma models, are provided. *P* values were adjusted for multiple comparisons across patterns including 'others' using the Benjamini–Hochberg method (adjusted *P* value for others, $1.1 \times 10^{-4}$).

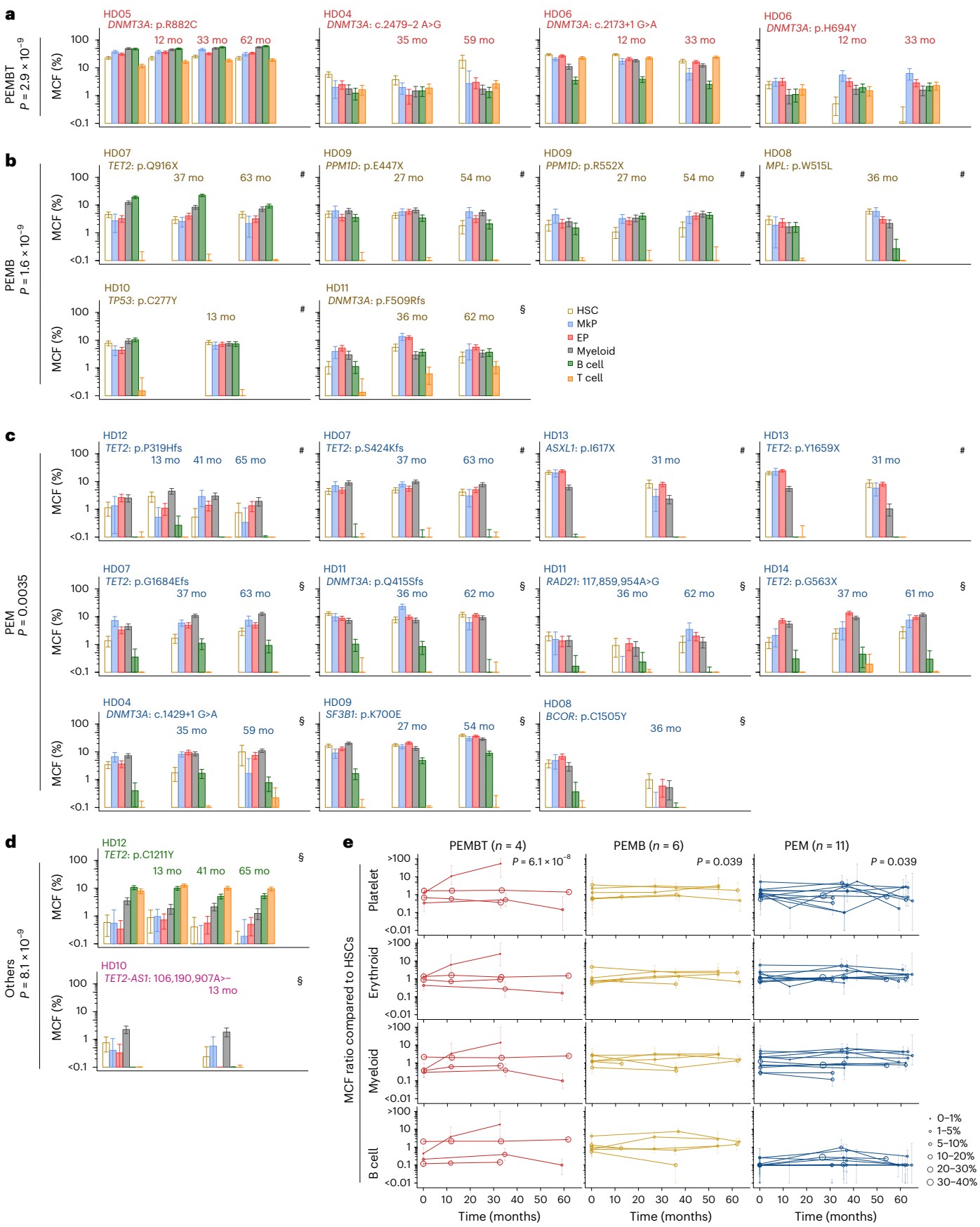

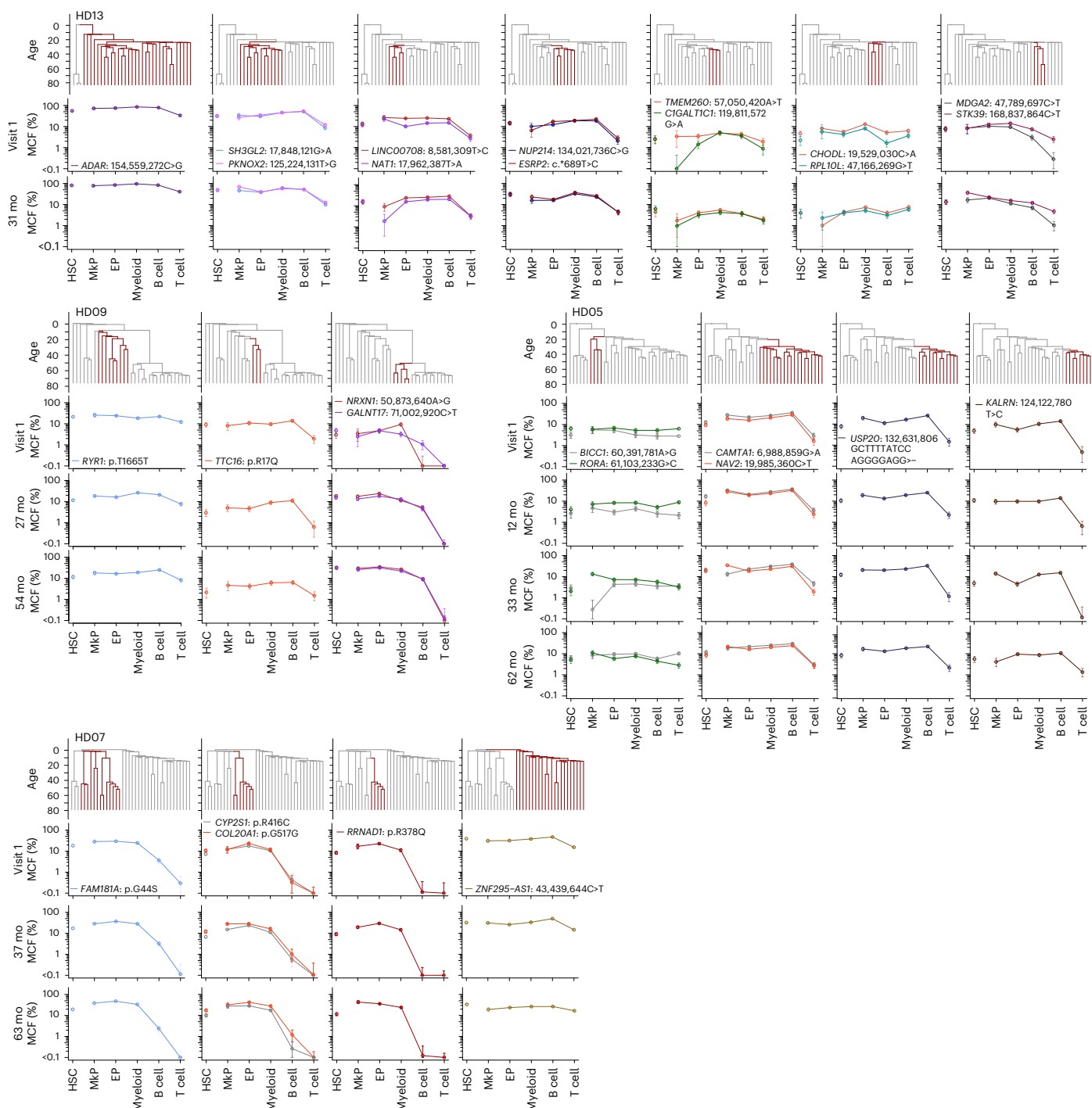

**Fig. 7 | Phylogenetic analysis infers stable lineage contribution over time of ancestral and descendant HSC clones.** MCFs in the different blood cell lineages for ancestral and descendant HSC clones marked by CH-US mutations, identified through single-colony WGS from serially collected BM samples from four donors (HDs 13, 09, 05 and 07; see Fig. 6 for serial analysis of the same BM samples of HSC clones marked by CH driver mutations) (error bars show 95% credible intervals estimated with the Bayes method taking into account the number of droplets and the number of input cells (see Methods for details)). Clades marked by each specified mutation on the phylogenetic tree are highlighted in red in the trees above each clonal plot. The $y$ axis in the trees is scaled based on donor age, starting (top of the phylogenetic tree) at the time of conception (40 weeks before birth) and ending (bottom of the phylogenetic tree) at the time of collection of the investigated BM sample. The correlation between clone sizes in serial samples and those in the initial-visit samples was assessed using mixed gamma models, and statistical significance was assessed using two-sided $z$-tests, yielding a $P$ value of $3.70 \times 10^{-21}$.

experimentally achieves the HSC clonal expansion that occurs normally in aged humans. While the main limitation of this approach is the potential impact of the transplantation setting, it obviates the main limitations of studying expanded HSC clones in aged healthy human participants, namely the potential impact of the clonal mutations and being applicable primarily to analysis in aged individuals.

In light of these differences in experimental settings, it is striking how the steady-state aged human HSC lineage replenishment and restriction and bias patterns (PEMBT, PEMB and PEM) observed here as well as how their relative prevalence changes with age closely match the main lineage replenishment patterns of transplanted single HSCs from young adult as well as old mice[15,16,48]. The only exception to this is the

platelet-restricted and -biased HSCs observed in mice[15,49], which would probably not be detected through the mutational screening performed in the current human study using BM MNCs in which megakaryocytes are exceptionally rare when compared to the other white blood cell lineages. The highly conserved HSC clonal lineage replenishment patterns observed between mice and humans, despite being assessed in very different experimental settings, are likely explained by our finding that these lineage replenishment patterns are very stable over time and upon transplantation, compatible with an intrinsically tightly regulated mechanism. Epimutations have recently been reported to provide lineage-tracing capabilities comparable to those of somatic nuclear mutations, enabling future simultaneous assessment of molecular cues governing HSC lineage restriction and bias[50].

The significance of the here identified PEMB- and PEM-restricted and -biased human HSCs and their stable clonal contribution to steady-state hematopoiesis alongside fully multilineage HSCs warrants further investigations, including whether they respond differently to specific blood lineage losses or hematopoietic stress as well as their roles as reservoirs for genomic alterations and the origin of hematological malignancies with distinct lineage affiliations.

## Online content

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

## Methods

### Ethical statement

Our study complies with all relevant ethical regulations and was approved by the institutional review boards at Karolinska Institutet. All human samples were collected with informed consent approved by the Stockholm regional ethical review board (EPN 2018/901-31). All mouse experiments were performed according to institutional guidelines, and we obtained permissions from the ethics committees at Stockholms Djurförsöksetiska Nämnd (17978-18 with amendments 18539-21).

### Healthy bone marrow donors and experimental design

BM and buccal swabs from 93 healthy individuals with rigorous inclusion criteria ensuring the absence of current or previous blood disorders or cancer and without ongoing treatment with immunosuppressive or other drugs likely to affect hematopoiesis were collected. Peripheral blood was analyzed to confirm that blood parameters were within the normal reference range and without any evidence of previous or current hematological disease (Extended Data Fig. 2b). For 11 individuals, more than one BM sample was collected 12–65 months after the initial sample (Extended Data Figs. 1 and 2c and Supplementary Table 1). BM MNCs were isolated using Lymphoprep (Axis-Shield), viably frozen in FCS (Sigma-Aldrich) medium supplemented with 10% DMSO (Sigma) and stored in liquid nitrogen until they were subjected to further analysis. DNA was extracted from buccal swabs and from more than one million BM MNCs using the QIAamp DNA Blood Mini Kit (Qiagen) or the Maxwell RSC Cultured Cells DNA Kit (Promega) according to the manufacturer's instructions.

The overall experimental design is summarized in Extended Data Fig. 1, and the specific experiments conducted for each BM sample are detailed in Supplementary Table 1. BM MNC DNA from the initial visit of all 93 HDs was subjected to ECTS to screen for CH mutations. Among the 71 donors found to harbor CH mutations, 33 of the 34 individuals with clones of sufficient size (defined as MCF ≥ 2% for CH driver mutations and MCF ≥ 1% for CH-US mutations) were further analyzed for lineage contribution using ddPCR on purified HSPCs and mature blood cells isolated from the initial BM samples. One donor was excluded due to a lack of viable cells. Serial dynamics were assessed in 11 donors using longitudinally collected BM samples. Single-cell colony WGS was conducted for ten of the 33 donors in whom PEMBT or PEM clones of adequate size were identified. For four of these ten donors, selected clones identified through single-colony WGS (either lacking CH driver mutations or ancestors or descendants of CH clones) were also analyzed longitudinally. Finally, xenotransplantation into immunodeficient mice was performed for three donors in whom PEM-restricted clones were detected. Definitions of PEMBT and PEM restriction and bias are provided in Assessment and classification of clonal HSC lineage contribution patterns.

### Error-corrected targeted DNA capture sequencing analysis

BM MNC DNA isolated from all 93 HDs was subjected to ECTS for identification of somatic mutations targeted to 23 genes encompassing the most recurrently mutated genes reported in CH[10,18,23,36,51,52]. Hybridization probes were designed to capture the entire exonic regions of eight genes (PPM1D, DNMT3A, TET2, ASXL1, TP53, CBL, RAD21 and BCOR) and specific mutation hotspot regions for 15 genes (SRSF2, SF3B1, U2AF1, JAK2, MPL, KRAS, NRAS, HRAS, KIT, BRAF, FLT3, IDH1, IDH2, GNAS and MYD88) (Supplementary Table 4) to capture >95% of the mutations in previously reported individuals with CH[10,18,23,36,51,52].

Briefly, 200 ng (equivalent to 30,000 cells) of DNA fragmented with an E220 Focused-ultrasonicator (Covaris) was subjected to ECTS library preparation using a KAPA HyperPrep Kit (Kapa Biosystems), xGen dual-index unique molecular identifier (UMI) adaptors (IDT) and custom-designed capture probes targeting the listed 23 genes (xGen custom probe, IDT). After hybridization using an xGen

Hybridization and Wash Kit (IDT), the enriched fragments were amplified using KAPA HiFi HotStart ReadyMix (Kapa Biosystems). The prepared libraries were sequenced on the NovaSeq 6000 system (Illumina) at the National Genomics Infrastructure, hosted by the Science for Life Laboratory in Stockholm. Sequencing reads were mapped to GRCh37 using the Burrows–Wheeler Aligner (v.0.7.17). After grouping reads with the same UMI using Picard (v.2.20.2) (http://broadinstitute.github.io/picard) and fgbio (v.0.8.1) (https://github.com/fulcrum-genomics/fgbio), consensus reads were generated using fgbio FilterConsensusReads with the following options: min-read = 5, min-base-quality = 40, max-read-error-rate = 0.025, max-base-error-rate = 0.1 and max-no-call-fraction = 0.05. Consensus reads were subjected to indel realignment and base quality score recalibration using GATK3 (ref. 53) (v.3.8) and recalculation of mismatching positions and/or edit distance to the reference (MD/NM) tags using SAMtools (v.1.9), resulting in a median of 5,037 depth per sample (range of 3,145–9,177 depth) with unique UMI in the final deduplicated BAM files from BM MNCs (n = 93). DNA from 24 buccal swab samples from 23 donors (including one technical replicate from donor HD66) was used as the germline control for the removal of SNPs and sequencing errors and sequenced with a median depth of 4,082 (2,162–9,291). Mutation calling was performed using EBCall[54] (https://github.com/friend1ws/EBCall), and mutations were annotated using ANNOVAR[55] (v.8 June 2020). Occurrence of mutations was evaluated using COSMIC[56] v.91. After mutation calling, read-based variant allele frequencies (VAFs) were calculated based on the reads using the GenomonMutationFilter (v.0.2.8) (https://github.com/Genomon-Project/GenomonMutationFilter), See Supplementary Notes for parameters for mutation calling and filtering.

Mutations annotated as 'synonymous SNV', 'intronic' or 'ncRNA_intronic' by ANNOVAR were treated as CH-US mutations. The positive predictive value and the sensitivity of the ECTS were assessed by mixing DNA from Jurkat cells with DNA from K562 cells (both from ATCC) at different ratios (0.1%, 0.2% and 1%) and have been described and reported previously[20].

As whole-exome sequencing of BM MNCs isolated from 20 healthy older (≥71 years of age) individuals (Supplementary Table 1) did not reveal any copy number alterations in regions targeted by ECTS, MCF reflecting cellular clonal involvement was calculated based on 2 × VAF for all mutations except mutations in male individuals located on the X chromosome where MCF = VAF.

### Whole-exome sequencing

Whole-exon sequencing was performed on 20 donors (Supplementary Table 1), primarily to check for copy number alterations as described in Error-corrected targeted DNA capture sequencing analysis. BM MNC DNA isolated from samples taken on the first visit was subjected to library preparation for bulk whole-exome sequencing using a Lotus DNA Library Prep Kit (IDT) and the xGen Exome Hyb Panel v2 (IDT). Hybridization and post-capture PCR were performed as described for ECTS followed by sequencing using the NovaSeq 6000 system (Illumina) at the National Genomics Infrastructure in Stockholm. Paired buccal swab DNA was used for excluding SNPs as well as sequencing errors. Sequencing reads were aligned to the human genome reference (GRCh37) using the Burrows–Wheeler Aligner (v.0.7.17) with default parameter settings. PCR duplicates were marked using Biobambam (v.2.0.87). Errors associated with enzymatic fragmentation were removed using FADE (v.0.2.2)[57]. BM MNC and paired buccal swab DNA libraries were sequenced with a median of 303 (range, 132–716) and 268 (18–637) coverage after deduplication, respectively. Mutation calling was performed using GenomonFisher (v.0.4.4) (https://github.com/Genomon-Project/GenomonFisher). See Supplementary Notes for parameters for mutation calling and filtering.

Copy number analysis was performed using CNACS[58] (https://github.com/OgawaLabTumPath/CNACS).

## Flow cytometric purification of hematopoietic stem and progenitor cells and mature blood cell lineages

Viably frozen BM MNCs were thawed in a water bath at 37 °C and washed with Dulbecco's PBS (Gibco) supplemented with 20% FCS (Sigma-Aldrich) and 100 µg ml⁻¹ DNase I, Bovine Pancreas (Sigma-Aldrich). After a 5-min incubation with human FcR Blocking Reagent (Miltenyi Biotec), cells were stained with fluorescently conjugated monoclonal antibodies for 15 min at 4 °C for identification of different HSPCs[25,26] and mature cells. See Supplementary Table 5 for antibody details. Stained cells were washed with PBS supplemented with 5% FCS and 2 mM EDTA (Invitrogen) with DAPI (Invitrogen) included for the HSPC panel and 7AAD (Sigma-Aldrich) for the mature cell panel for identifying live cells immediately before flow cytometry analysis. HSPC subsets were sorted on a FACSAria Fusion instrument (BD Biosciences) using FACSDiva software (v.8.0.2) and included fluorescence-minus-one and single-stained controls. Data were subsequently analyzed with FlowJo (BD, v.10.10.0). The following cell populations were sorted using these panels: HSCs, Lineage⁻CD34⁺CD38⁻CD90⁺CD45RA⁻; MkPs, Lineage⁻CD34⁺CD38⁺CD41a⁺; EPs, Lineage⁻CD34⁺CD38⁺CD41a⁻CD123⁻CD45RA⁻; pro-B cells, Lineage⁻CD34⁺CD19⁺; myeloid cells, CD14⁺CD33⁺CD3⁻CD19⁻CD56⁻; B cells, CD19⁺CD3⁻CD33⁻CD56⁻; T cells, CD3⁺CD8a⁺CD4⁻CD19⁻CD33⁻CD56⁻ and CD3⁺CD4⁺CD8a⁻CD19⁻CD33⁻CD56⁻.

Lineage markers include CD2*, CD3, CD4, CD7*, CD8a, CD10*, CD11b, CD14, CD19*, CD20, CD56 and CD235ab (* anti-CD2, anti-CD7, anti-CD10 and anti-CD19 antibodies were removed from the Lineage panel for identification of pro-B cells).

## Whole-genome DNA amplification of purified cell populations

Cells purified by fluorescence-activated cell sorting were subjected to whole-genome DNA amplification using a REPLI-g Single Cell Kit (Qiagen) according to the manufacturer's instructions. Briefly, 112–2,003 cells per population were sorted directly into 0.2-ml PCR tubes containing 3 µl Buffer D2 and incubated for 10 min at 65 °C. After adding 3 µl Stop Solution, cells were subjected to whole-genome amplification by adding 40 µl REPLI-g sc master mix to the samples and incubating at 30 °C for 8 h. Amplified DNA was subjected to two rounds of purification with 1.8× volume of AMPure XP Beads (Beckman Coulter) before the final amplified product was dissolved in TE buffer.

## Droplet digital PCR

DNA from the purified cells was subjected to highly sensitive and quantitative ddPCR analysis after whole-genome amplification as previously described[25] for mutations identified by ECTS or WGS. In phylogenetic analysis for mutations identified by WGS, only those showing ≥2% MCFs in Lineage⁻CD34⁺CD38⁻CD90⁺CD45RA⁻ HSCs were subjected to analysis (Extended Data Figs. 6–8). A 20-µl PCR reaction containing 1× ddPCR Supermix for probes without dUTP (Bio-Rad), 1× mutation detection primer–probe (designed and ordered from Bio-Rad or IDT; Supplementary Table 6) and 60 ng template DNA, either isolated from >1 million BM MNCs or from cell populations purified by flow cytometry and subjected to whole-genome DNA amplification. Sample preparation and droplet generation were carried out according to the manufacturer's instructions using a QX200 Droplet Generator (Bio-Rad) as previously described[25]. Plates were read on a QX200 Droplet Reader (Bio-Rad) with QuantaSoft (v.1.7.4) (Bio-Rad), and the results were analyzed using QuantaSoft Analysis Pro (v.1.0.596) (Bio-Rad). Genomic DNA samples from bulk BM MNCs from the same donor and peripheral blood from anonymized donors were used as positive and negative controls, respectively. REPLI-g no-template amplified controls were also included in each ddPCR plate. All samples generated more than 8,000 accepted events. The two-dimensional fluorescence amplitude for each well was visually inspected, and samples with abnormal signal intensities were excluded from further analysis.

Mean MCFs and their credible intervals were estimated through Bayesian inference using CmdStan (v.2.33.1) based on the number of droplets assigned to the two-dimensional ddPCR quadrants (double negative, single positive either for mutant or wild type, and double positive), the sex of the individual, the chromosomal location of the mutation and the number of sorted cells subjected to REPLI-g amplification. Briefly, the MCF in the original samples was modeled as a beta distribution, taking into account sex and chromosome type (autosome versus sex chromosome). Extraction efficiency (defined as the efficiency with which targeted DNA molecules are successfully partitioned into individual ddPCR droplets) was modeled using an exponential distribution, reflecting the fact that each droplet typically contains zero or only a few target molecules. The likelihood contributions from droplets were modeled using Poisson distributions, with $\lambda$ parameters representing the expected number of wild-type and mutant molecules per droplet. Parameter estimation was performed using Markov chain Monte Carlo methods implemented in CmdStanR (v.0.6.1). Additional details, including the specific prior distributions, model code and convergence diagnostics, are available through the Swedish National Data Service Repository (https://doi.org/10.48723/vany-1s63) (ref. 59). For ddPCR data generated from bulk BM MNC DNA, MCF was estimated using 9,090 cells as the parameter for cell number, representing the predicted number of cells corresponding to 60 ng DNA, as a single cell contains approximately 6.6 pg of DNA ($3 \times 10^9$ bp (haploid) × 2 (diploid) × 660 Da per bp × $1.67 \times 10^{-12}$ pg per Da). MCFs from ECTS and those from ddPCR in BM MNCs showed high concordance, supporting the validity of these two platforms (Supplementary Fig. 1a).

A sample was defined as positive for the analyzed mutation if the overlapped area of posterior distributions by Bayesian inference between the targeted sample and the normal control was <5%. A sample was defined as likely positive, inconclusive or negative if the overlapped area was ≥5% and <20%, ≥20% and <40%, or ≥40%, respectively. Positive or likely positive and negative or likely negative were treated as positive and negative, respectively, in Figs. 2a and 5e–i. The number of cells sorted, the number of ddPCR events observed and the performance of the probes are parameters affecting the sensitivity of the ddPCR analysis, and therefore the cutoff varies depending on the sample. As a result, the lowest detected positive value was 0.12% (Supplementary Fig. 1b).

For mutations identified by ECTS in the first-visit samples, 51 CH driver mutations with ≥2% MCFs and ten CH-US mutations with ≥1% MCFs from the donors with a sufficient number of frozen BM MNCs were selected for further ddPCR analysis. The estimated number of mutant cells in samples assessed for the different cell lineages is provided in Supplementary Table 7.

## Assessment and classification of clonal HSC lineage contribution patterns

The lineage contribution of individual HSC clones was assessed for the 57 mutations (47 CH driver and ten CH-US mutations) found to reliably originate in HSCs (among the 61 mutations subjected to ddPCR). Considering that multipotent HSCs have the potential to replenish all five major blood cell lineages investigated (PEMBT), we first assessed to what degree individual HSC clones contributed to all five lineages (PEMBT; multilineage) or whether they were lineage restricted, defined as lacking detectable contribution to one or more of the five lineages. We also assessed whether individual HSC clones (whether multilineage or lineage restricted) had a considerable lineage bias. To assess this, clonally involved lineages were, for each individual HSC clone, arranged according to their MCF, from highest to lowest. Subsequently, the ratio of MCFs for the most and second-most clonally involved lineage was calculated and then for the second and third etc., until MCF ratios were obtained for all nearest-neighbor comparisons. Only if the MCFs of the nearest neighbor showed more than fivefold differences was the HSC clone considered to have a considerable lineage bias between those lineages. To be defined as being biased for more than one lineage, each

of the lineages positively biased in the HSC clone should have more than fivefold higher MCF than those for all lineages with negatively bias in the clone. Moreover, to define an HSC clone to be positively biased toward a specific combination of lineages (that is, PEM or PEMB) or for a fully multilineage (PEMBT) HSC clone to be considered lineage balanced, the ratio between the lineages with the highest and lowest MCFs should be less than ten, as documented in Supplementary Fig 1c.

To assess to what degree observed lineage restriction and lineage bias patterns of individual HSC clones fluctuated or were stable over time, we compared the lineage contribution patterns of 22 individual HSC clones (in 11 donors) in sequential BM samples obtained with an interval of 12–65 months. Statistical assessment was conducted as described in Statistical analysis and reproducibility.

## In vitro colony expansion of single hematopoietic stem and progenitor cells

Single HSPCs were identified by flow cytometry as described above and deposited with an automated cell deposition unit directly into individual wells of a 96-well U-bottom plate containing 100 μl StemSpan SFEM medium (Stemcell Technologies) supplemented with 10% FCS (Sigma), $10^{-4}$ M β-mercaptoethanol (Sigma), 1% L-glutamic acid (Gibco), 1% penicillin–streptomycin (PAA Labs), 10 ng ml$^{-1}$ human thrombopoietin (PeproTech), 10 ng ml$^{-1}$ human SCF (StemGen), 10 ng ml$^{-1}$ human FLT3 ligand (Immunex), 5 ng ml$^{-1}$ human IL-3 (PeproTech), 10 ng ml$^{-1}$ human GM-CSF (Berlex), 10 ng ml$^{-1}$ human G-CSF (Amgen) and 1 IU ml$^{-1}$ human erythropoietin (Roche). Index sorting data obtained with FAC-SDiva (v.8.0.2) were analyzed with FlowJo (v.10.10.0) followed by the R package flowCore (v.2.10.0). Individual clones were expanded for up to 5 weeks, collected and subjected to DNA isolation using the Maxwell RSC Cell DNA Purification Kit (Promega). Before DNA isolation, a small aliquot was taken from the colony cell suspension and subjected to whole-genome amplification using the REPLI-g Single Cell Kit (Qiagen) as described above but at one-half to one-third of the described volume. To aid the selection of colonies for downstream WGS, amplified colony DNA was subjected to ddPCR or Biomark HD (Fluidigm) genotyping for mutations detected by ECTS, whole-exome sequencing and/or single-colony WGS (Supplementary Table 8). Only colonies with a VAF near 50% were subjected to single-colony sequencing to exclude those potentially coming from wells containing more than one cell.

## Single-cell-derived colony WGS analysis

Non-amplified DNA extracted from 354 genotyped single-cell-derived colonies using the Maxwell RSC Cell DNA Purification Kit (Promega) from ten donors was subjected to WGS library preparation using the Lotus DNA Library Prep Kit (IDT) or the xGen DNA Library Prep Kit EZ (IDT), followed by sequencing using the NovaSeq 6000 system (Illumina) at the National Genomics Infrastructure in Stockholm with an average coverage of 13.91 depth per sample (8.13–30.71). Paired buccal swab DNA was used for excluding SNPs as well as sequencing errors. Processing of sequencing data and mutation calling were performed as for whole-exome sequencing data. Called mutations were annotated using ANNOVAR (v.8 June 2020)[55]. See Supplementary Notes for parameters for mutation calling and filtering.

The *DNMT3A* p.R598X mutation in HD24 and *DNMT3A* p.F731S, and *SF3B1* p.M757T in HD65, which were identified by ECTS, were not initially detected in single-colony WGS due to lower coverage in these regions. These mutations were manually validated and rescued after their presence was confirmed through inspection using the Integrative Genomics Viewer. We excluded 21 colonies that showed low depth or skewed VAF peaks in detected mutations and confirmed the single-cell origin of colonies through normalized VAF distribution around 50% in the remaining 333 colonies analyzed (mean number of colonies per donor, 33.3 (minimum–maximum, 16–57)) (Supplementary Fig. 2).

Colonies derived from the same BM sample from HD03 were sequenced on two separate occasions (several years apart), and mutations shared among colonies sequenced on the later occasion were observed that were absent in the earlier sequencing run. These mutations were therefore presumed to be errors induced through long storage and were consequently excluded. Because these errors could lead to inaccurate age estimation, HD03 was also omitted from the statistical analysis for age dating of HSC clones.

Structural variants were identified with GenomonSV (https://github.com/Genomon-Project/GenomonSV). See Supplementary Notes for parameters for calling and filtering.

In total, 69 structural variants were observed in 73 colonies from nine donors. We evaluated each structural variation for the presence of breakpoints on the driver gene sets listed in Definition of driver mutations for phylogenetic analysis. The breakpoints of two structural variations from two donors (HD07 and HD13) involved cancer, myeloid malignancy or CH-related genes, but none of them were observed in multiple colonies (Supplementary Fig. 3 and Supplementary Table 9).

Copy number analysis of WGS data was performed using Control-FREEC v.11.6 (ref. 60) and the ASCAT R package v.3.1.1 (ref. 61). Chromosome Y loss in male donors was evaluated by retrieving sequence depth information of chromosomes X and Y from mapped BAM files using the idxstats function in SAMtools (v.1.9). Deletion of 5q and uniparental disomy in 3q in a colony from HD02 and a colony from HD03, respectively, were the only identified chromosomal abnormalities in all colonies (Supplementary Fig. 3).

Mutational signature analysis for single-nucleotide mutations detected by single-colony WGS was performed using MutationalPatterns (v.3.7.0)[62]. Optimal nonnegative linear combination of mutation signatures to reconstruct the mutation matrix was evaluated by fitting to the combined reference of COSMIC Mutational Signatures v.2 (https://cancer.sanger.ac.uk/signatures/signatures_v2/) and single-base substitution (SBS)blood[63]. The similarity of 96 trinucleotide signatures was evaluated using cosine similarity and was clustered with the ward.D2 algorithm using Euclidean distances. This mutation pattern analysis confirmed that the major source of mutations is SBSblood[63], which is compatible with previous reports[13,20] (Supplementary Fig. 4).

## Reconstruction of phylogenetic clonal trees based on mutations identified in single-cell-derived colonies

VAFs of identified mutations were calculated using GenomonMutationFilter (v.0.2.8) (https://github.com/Genomon-Project/GenomonMutationFilter) in all the colonies and the buccal swab DNA derived from the same donor. The binary mutational matrix of colonies and buccal swab DNA in each donor was created with the following criteria (VAF ≥ 0.15 and mutant reads ≥ 2, positive; mutant reads = 0 and ≥5 total reads, negative; and the rest, unknown) and was converted into the PHYLIP format. Consensus phylogenetic trees were reconstructed using MPBoot[64] (https://github.com/diepthihoang/mpboot) with 1,000 iterations. Using SNP sensitivity information (see Supplementary Notes for parameters for the methods of SNP sensitivity calculation), we corrected the length of branches using the get_corrected_tree R function[6] downloaded from https://github.com/emily-mitchell/normal_haematopoiesis/tree/main. Mutations were allocated to branches using the R package treemut (v.1.1). For calculating the timing of mutation acquisition, the mutation rate was assumed to be constant throughout life. Phylogenetic trees were converted to an age scale using the functions in https://github.com/emily-mitchell/normal_haematopoiesis, assuming 40 weeks from conception to birth, to assess the age at which different somatic mutations were acquired. We estimated the 95% CI for the age of acquisition of CH driver mutations as follows: The mutation count along each branch was used to calculate the 95% CI using a Poisson distribution, where $\lambda$ was set to the average mutation rate of all the colonies for each donor. To estimate the lower bound of the age, we assumed lower mutation counts on upstream branches and higher mutation counts on downstream branches. Conversely, to estimate

the upper bound, we assumed higher mutation counts on upstream branches and lower mutation counts on downstream branches. Both resulting trees corresponding to the upper and lower bounds of the 95% CI were then transformed into ultrametric trees by scaling branch lengths to match the sample age.

Expanded clone branches were identified from the phylogenetic tree, and several mutations on those branches where ddPCR probe design was feasible were randomly selected for ddPCR analysis. Except for clones identified by ECTS, only clones with an MCF ≥ 2% in HSCs were included in the analysis (Figs. 3, 5 and 7 and Extended Data Figs. 6–8). When ddPCR results for two or more mutations on the same branch were available, we prioritized CH driver mutations, followed by the mutation with the highest MCF in HSCs for the analysis (Fig. 5e–j).

Based on the following observations and rationale, the age at the distal end of the phylogenetic tree branch where the marker mutation was acquired was treated as the starting point of clonal expansion[65] and used for statistical analysis:

- CH driver mutations were frequently observed in clades with long branches, suggesting that these mutations likely drive clonal expansion and were acquired toward the distal end of the branches.
- When branch length was short, the age difference between the distal and proximal ends was minimal.
- In most cases, ddPCR analysis of multiple mutations within the same branch revealed consistent contribution patterns, indicating that the position along the branch had little impact, presumably because descendants had replaced their ancestors (Extended Data Figs. 6–8).

However, as other CH-US mutations can occur at any point along a branch, their timing will be more uncertain than for CH driver mutations, enhancing the possibility of subclonal heterogeneity within the same branch. Therefore, we also tested alternative scenarios in which other CH-US mutations were assumed to have been acquired at the midpoint or at the proximal end of the branch while keeping the acquisition time of CH driver mutations fixed at the distal end. Importantly, these alternative assumptions yielded similar results with regard to the timing of the occurrence of identified mutations (Supplementary Fig. 5).

The association between $\log_{10}$-transformed ratios of clonal mature lineages and the age at which mutations were acquired in the phylogenetic trees was evaluated using a linear mixed model, implemented with the lmer function in the R package lme4.

To assess whether sibling clones (defined as subclones sharing the same upstream ancestral mutation) at the distal ends of phylogenetic trees exhibited significantly concordant lineage contributions, we conducted a permutation test based on the most frequent lineage pattern observed within each clade originating from the earliest ancestral clone. For each clade containing ≥2 terminal clones that exhibited one of the following patterns (positive in all lineages, negative in T cells or negative in both B and T cells), we calculated the proportion of clones exhibiting the most frequent pattern, referred to as the lineage concordance score. To evaluate the statistical significance of this observed score, we generated a null distribution by randomly sampling the same number of clones from these clades from all the donors 10,000 times. For each permutation, we calculated the maximum frequency of any lineage positivity patterns in the sampled set, yielding an empirical distribution of lineage concordance scores under the null hypothesis of no clade-specific bias. The empirical P value was defined as the proportion of permutations in which the randomized score was equal to or greater than the observed score.

We also performed a spatial autocorrelation analysis of lineage contributions using Moran's I-test[66] based on phylogenetic distances. Because the number of analyzed clades per donor was relatively small, we pooled all donor data into a single analysis and assigned a weight of zero to clade pairs from different individuals. MCFs of five lineages (PEMBT) were normalized to those of HSCs, and a weight matrix was generated based on phylogenetic age distances, defined as the inverse of one plus the distance (that is, 1/(1 + distance)). Diagonal entries of the matrix (that is, self-comparisons) were set to zero, as Moran's I evaluates similarity between neighboring units and excludes self-correlation by definition. These normalized values were then analyzed using Moran's I-test with the specified weight matrix, implemented via the R package spdep under a one-sided alternative hypothesis (greater) to assess positive spatial autocorrelation.

### Definition of driver mutations for phylogenetic analysis

Nonsynonymous mutations in exon regions or mutations on splicing sites in 736 genes registered in the COSMIC Cancer Gene Census[67] (https://cancer.sanger.ac.uk/census, accessed on 6 March 2023) and 95 previously defined myeloid genes[6] were considered driver mutations. These candidate driver genes were annotated on the phylogenetic trees together with 17 CH genes (*DNMT3A*, *TET2*, *ASXL1*, *SRSF2*, *U2AF1*, *PPM1D*, *SF3B1*, *TP53*, *JAK2*, *CBL*, *KRAS*, *GNB1*, *CTCF*, *BRCC3*, *IDH1*, *PTPN11* and *IDH2*) and 18 genes (*ZNF318*, *HIST2H3D* (official gene name: *H3C13*), *SPRED2*, *MTA2*, *YLPM1*, *ZBTB33*, *ZNF234*, *SRSF1*, *IGLL5*, *MYD88*, *SIK3*, *CHEK2*, *CCDC115* (official gene name: *H3C13VMA22*) *BAX*, *SRCAP*, *SH2B3*, *CCL22* and *MAGEC3*) previously identified as genes under positive selection in HSCs from HDs[6,68].

### Transplantation into immune-compromised NSG mice

NSG mice were purchased from Jackson Laboratory and kept in a specific pathogen-free facility at the Karolinska Institute.

BM CD34$^+$ or CD34$^+$CD19$^-$ cells were obtained by CD34 magnetic bead enrichment according to the manufacturer's instructions (130-046-702, Miltenyi Biotec) or flow cytometry, respectively. An average of 28,269 (21,846–40,008) CD34$^+$ or CD34$^+$CD19$^-$ cells was intravenously injected via the tail vein into sublethally irradiated (1.25 Gy, X-ray source) 3–4-week-old female NSG mice (two to four mice per donor). BM was isolated from transplanted mice 10–14 weeks after transplantation and stained with the following fluorescently conjugated monoclonal antibodies. See Supplementary Table 5 for antibody details. Human myeloid cells (mTER119$^-$mCD41$^-$mCD45$^-$hCD235ab$^-$hCD45$^+$CD33$^+$CD19$^-$) and human B cells (mTER119$^-$mCD41$^-$mCD45$^-$hCD235ab$^-$hCD45$^+$CD33$^-$CD19$^+$) were sorted by flow cytometry and subjected to mutational ddPCR analysis following whole-genome amplification as described above.

### Statistical analysis and reproducibility

No statistical method was used to predetermine sample size. We excluded WGS data from 21 colonies that showed low depth or skewed VAF peaks in detected mutations. Single-colony WGS data from the same BM sample from HD03 were omitted from the statistical analysis for age dating of HSC clones as described in Single-cell-derived colony WGS analysis. No randomization or blinding was used in this study.

Statistical analyses were performed using R (v.4.2.2). The significance of the two categorical variables was evaluated using Fisher's exact test. For comparison of two groups, the Wilcoxon rank-sum test was used, unless otherwise noted. Correlations between two variables were assessed using Pearson's correlation coefficient assuming normal distribution, but this was not formally tested, and significance was evaluated using a Pearson correlation test. Adjusted $R^2$ values were calculated from linear regression models. All reported P values were obtained using two-sided tests, and the significance level was set at 0.05. P values for multiple testing were corrected using the Benjamini–Hochberg method.

To assess the stability of lineage clonal size and mature cell productivity of HSCs for clones identified through ECTS, we evaluated the correlation of clone size and clonal ratio relative to HSCs in serial samples with those in the initial-visit samples using a mixed gamma model with a log link function, implemented in the glmmTMB package

in R. Mutation identifier, mature cell lineage and time (d) after the initial sample collection were included as random effects in both models. The clonal stability in serially collected samples was assessed for each lineage contribution pattern separately as well as for all patterns combined. The same analysis was also performed assessing each blood lineage separately. For serial analysis of clones identified by WGS, all patterns combined were assessed.

To evaluate whether the contribution of clones to B cells and myeloid cells before and after transplantation was correlated, we constructed a binomial mixed-effect model. The positivity of B cells and myeloid cells before and after transplantation was fitted, with mouse and mutation identifiers included as random effects to account for both mutational and intermouse variability.

### Reporting summary
Further information on research design is available in the Nature Portfolio Reporting Summary linked to this article.

### Data availability
All DNA sequence data including ECTS and single-colony WGS have been deposited in the data repository of the Swedish National Data Service under controlled access (https://doi.org/10.48723/313d-dd68). Access is restricted in accordance with European Union General Data Protection Regulation to protect participant privacy, as the data contain identifiable genetic variants. Researchers may request access via the Researchdata.se page (https://researchdata.se/en). Access requests are processed immediately. Given that the requestor presents complete and correct documentation data access will be granted within five business days for legitimate research purposes. Data transfer time varies depending on data size. Sample information, detected mutation lists for ECTS, whole-exome sequencing and single-colony WGS, ddPCR results and results of transplantation into NSG mice are available at the Swedish National Data Service Repository (https://doi.org/10.48723/vany-1s63)[59] with restricted access. Source data are provided with this paper.

### Code availability
Codes for data analysis and figure generation are openly available at the Swedish National Data Repository under a CC BY 4.0 license (https://doi.org/10.48723/vany-1s63)[59].

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

### Acknowledgements

Sequencing was performed by National Genomics Infrastructure Sweden and Science for Life Laboratory supported by the Swedish Research Council and the Knut and Alice Wallenberg Foundation. We are grateful to M. Tobiasson, J. Ungerstedt, C. Lindholm, J. Wiggh, P.A. Broliden, L. Fredriksson, M. Creignou and T. Gullberg for performing BM aspirations and the donors for their participation in the study. We acknowledge the use of the ddPCR instrument at CRISPR Functional Genomics, a SciLifeLab infrastructure unit and Karolinska core facility and the Karolinska animal experimental core facilities for assistance with handling and irradiation of mice. This work was supported by grants to S.E.W.J. from the Knut and Alice Wallenberg Foundation (KAW 2016.0105), the Tobias Foundation (4-1122/2014), the Torsten Söderberg Foundation, the Center for Innovative Medicine at Karolinska Institutet (613/06), the Swedish Research Council (538-2013-8995), the Swedish Cancer Society (23 3138 Pj) and the Medical Research Council (MC_UU_12009/5); to P.S.W. from the Swedish Research Council (2015-03561), the Swedish Cancer Society (22 2178 Pj), the Knut and Alice Wallenberg Foundation (Academy fellowship award, 2015.0195) and Radiumhemmets Forskningsfonder (224132 and 244282); and to T.Y. from the Dr Åke Olsson Foundation for Hematology Research (2021-00087). T.Y. was supported in part by the Fellowship of the Astellas Foundation for Research on Metabolic Disorders and the Mochida Memorial Foundation for Medical and Pharmaceutical Research.

### Author contributions

T.Y. conceptualized the study, designed, performed and analyzed experiments, performed computational analysis and wrote the paper. W.W.K. contributed to data analysis and provided input on computational coding. C.N. managed the biobank and coordinated BM aspirations and clinical data collection. S.M., F.G., K.H. and E.M. contributed to flow cytometry. A.W. contributed

to genotyping of single-cell colonies and provided input for computational coding. A.A. contributed to ddPCR experiments. A.H., E.C., M.L. and M.S.B. performed in vivo NSG experiments. I.B., T.M.-B., G.W., M.J. and M.H.W. coordinated clinical data collection and processing, freezing and registration of BM samples in the biobank. E.H.-L. contributed to establishment and coordination of the biobank. P.S.W. designed, performed and analyzed experiments and wrote the paper. S.E.W.J. conceptualized the study, established, led and coordinated the biobank, designed and analyzed experiments and wrote the manuscript. All authors read and approved the final paper.

## Funding

## Competing interests

The authors declare no competing interests.

## Additional information

**Extended data** is available for this paper at https://doi.org/10.1038/s41588-025-02405-w.

**Correspondence and requests for materials** should be addressed to Tetsuichi Yoshizato or Sten Eirik W. Jacobsen.

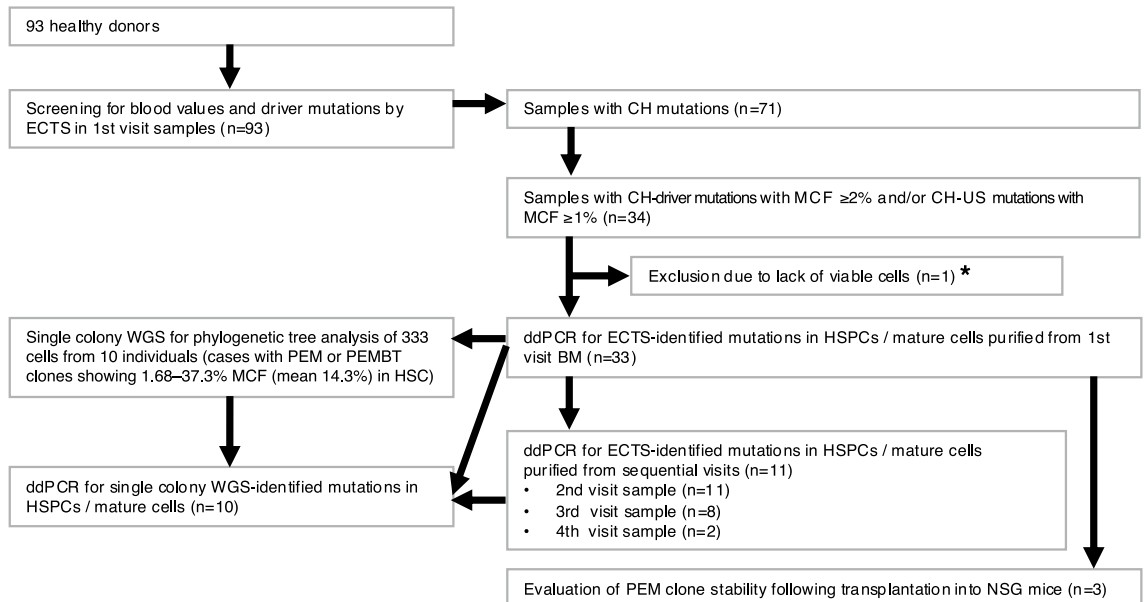

**Extended Data Fig. 1 | Consolidated Standards of Reporting Trials (CONSORT) diagram showing the study design.** Flow-chart showing processing and selection of bone marrow (BM) samples from 93 healthy donors for each analysis in the study. BM mononuclear cell (MNC) DNA from the initial visit of all 93 healthy donors was subjected to error-corrected targeted DNA sequencing (ECTS) to screen for clonal hematopoiesis (CH) mutations. Among the 71 donors found to harbor CH-mutations, 33 of the 34 individuals with clones of sufficient size (defined as mutant cell fraction (MCF) ≥ 2% for CH-driver and ≥1% for other CH mutations of undetermined significance (CH-US)) were further analyzed for lineage contribution using droplet digital PCR (ddPCR) on purified hematopoietic stem and progenitor cells (HSPCs) and mature blood cells isolated from the initial samples. One case was excluded due to a lack of viable cells (*). Serial dynamics were assessed in 11 donors using longitudinally collected samples. Single cell colony whole-genome sequencing (WGS) was conducted for 10 of the 33 cases in which PEMBT (platelet, erythroid, myeloid, B-cell, T-cell -balanced) or PEM (platelet, erythroid, myeloid -biased/restricted) CH clones with adequate size were identified (18 mutations; 1.68–37.2% (mean 14.3%) MCF in HSCs). For 4 of these 10 cases, selected clones identified through single colony WGS—either lacking CH-mutations or ancestors or descendants of CH clones—were also analyzed longitudinally. Finally, xenotransplantation into immunodeficient NOD.Cg-PrkdcscidIl2rgtm1Wjl/SzJ (NSG) mice was performed for 3 cases in which PEM-restricted clones were detected. See Supplementary Table 1 regarding which specific donor samples were used for which analysis.

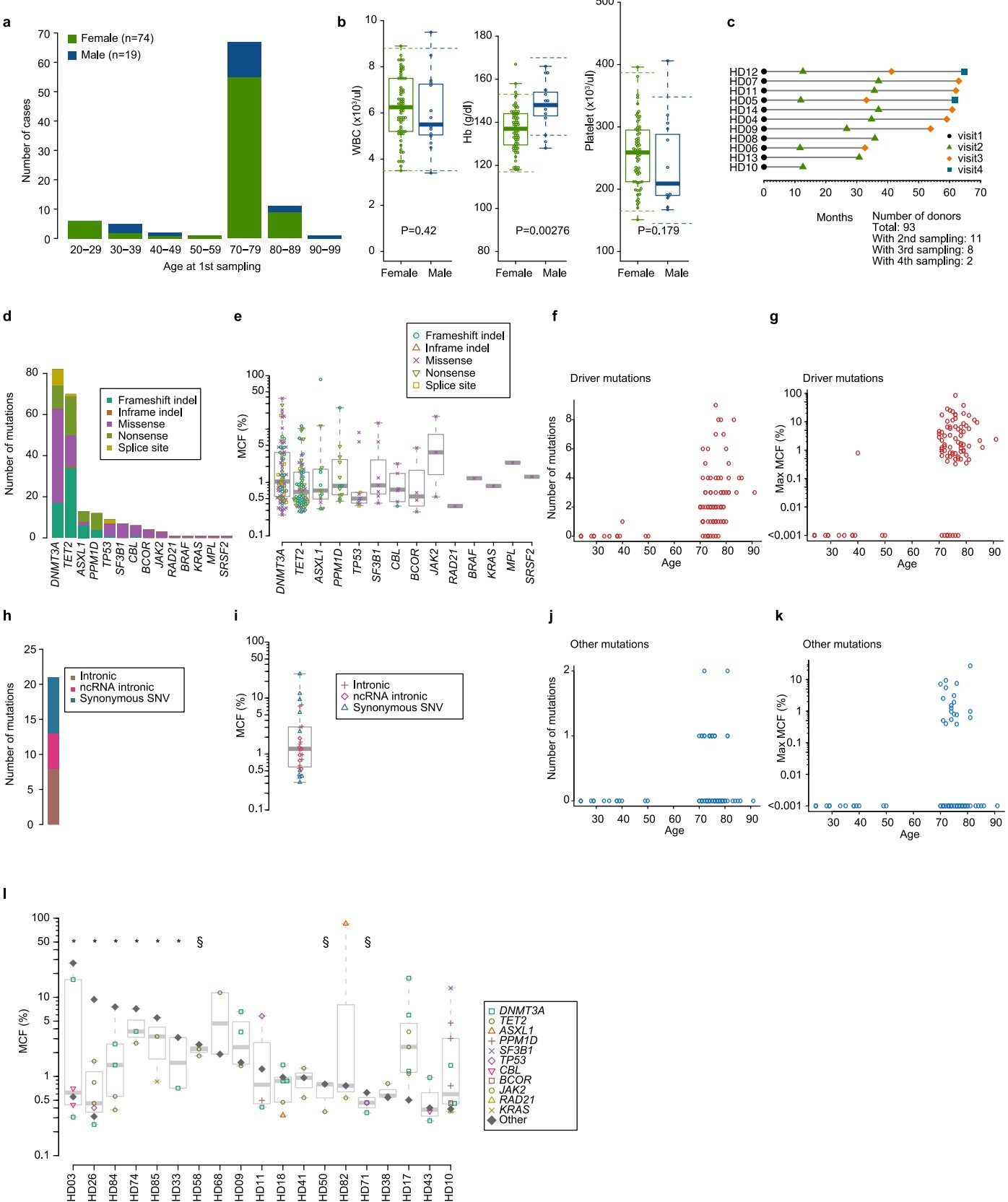

**Extended Data Fig. 2 | See next page for caption.**

**Extended Data Fig. 2 | Characterization of clonal hematopoiesis mutations in bone marrow of healthy donors. a.** Distribution of age (at first BM donation) and sex among recruited healthy BM donors. **b.** Peripheral blood white blood cell (WBC), hemoglobin (Hb), and platelet values at first BM collection. Each dot represents an individual healthy donor (HD) (female n = 64, male n = 15; Cases without blood count information were excluded (n = 14)). The center lines and boxes denote median values and the first and third quartiles, respectively. Whiskers indicate maximum and minimum values within 1.5× interquartile ranges. The green and blue horizontal dashed lines indicate the upper and lower reference limits for female and male donors, respectively. Statistical tests were performed using Wilcoxon rank sum test and P-values were adjusted by Benjamini–Hochberg method. **c.** Overview of number and timing of BM samples investigated for the different HDs in the study. The figure only illustrates cases with serial sample collection. The total number of cases (including those with single-point samples) and the number of cases with serial samples are indicated within the figure. **d.** Distribution of all detected driver mutations (VAF > 0.1%) by ECTS of DNA from BM MNCs isolated from HDs (n = 93). Types of mutations are indicated by different colors. **e.** Box plot showing clonal involvement based on percent MCF of detected driver mutations in BM MNCs shown in d. MCF estimation was performed based on the assumption that all mutations were heterozygous unless located on a sex chromosome in male donors. The center lines and boxes denote median values and the first and third quartiles, respectively. Whiskers indicate maximum and minimum values within 1.5× interquartile range. Types of mutations are indicated by different colors and marks. **f–g.** Number of driver mutations detected in BM MNCs (**f**) and MCF for the CH-driver mutations with the highest clonal involvement (**g**) in individual HDs based on age. Each circle indicates one HD. **h–k.** Corresponding data to panels **d**–**g** for other CH-US mutations (VAF > 0.1%). **l.** MCF distribution of detected CH-driver and CH-US mutations in each donor with identified CH-US mutations (n = 19). Cases are ordered by clone size of CH-US mutations. The center lines and boxes denote median values and the first and third quartiles, respectively. Whiskers indicate maximum and minimum values within 1.5× interquartile ranges. Targeted genes are indicated by different colors and marks. * and § indicate cases with maximum MCF of other mutations >1.5× and ≤1.5 & >1.0× of maximum MCF for identified driver mutations, respectively.

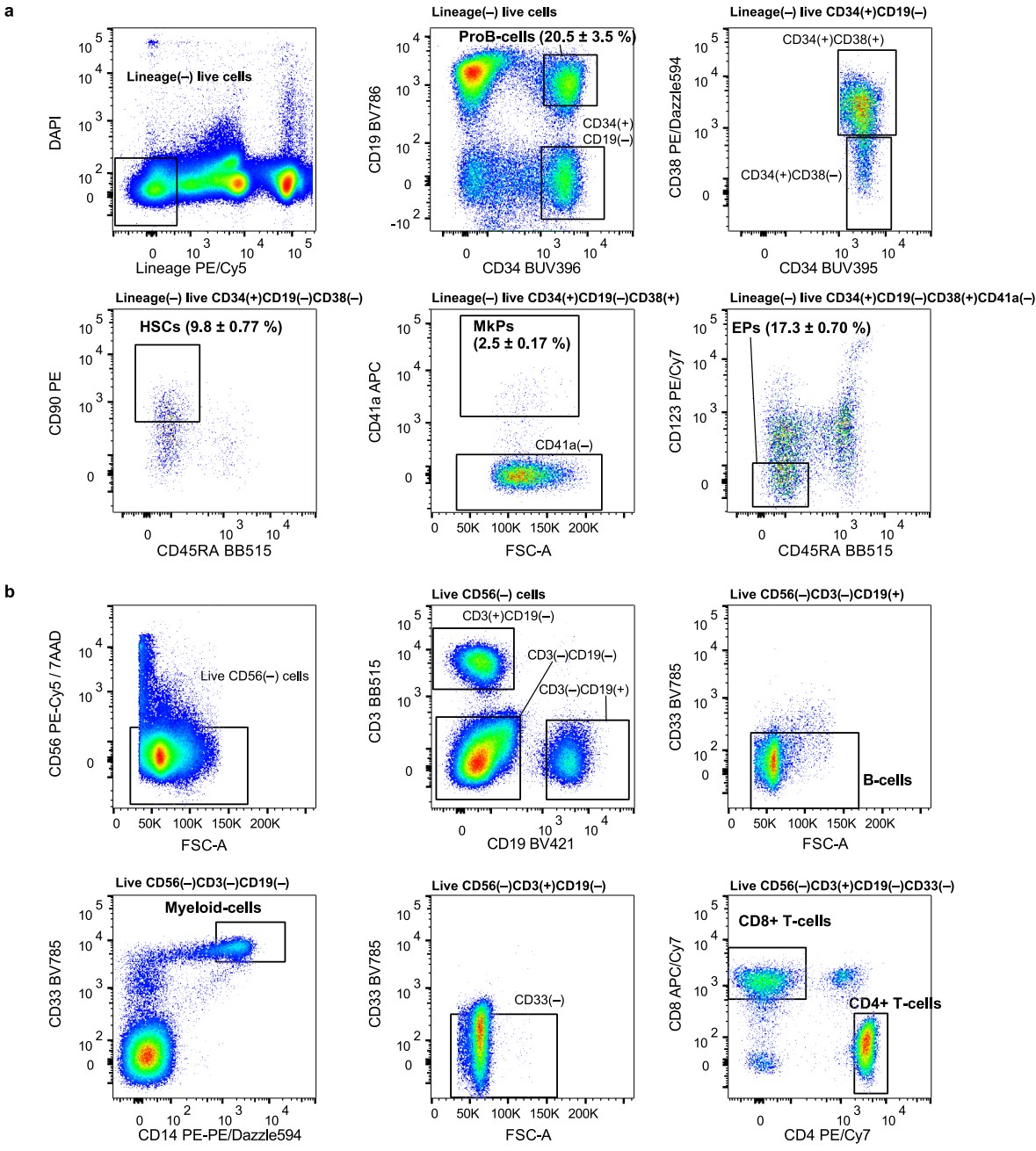

**Extended Data Fig. 3 | Identification and purification of hematopoietic stem and progenitor cells and mature blood cell lineages in healthy donors with clonal hematopoiesis. a.** Flow cytometry gating strategy for isolation of distinct HSPC populations in BM from HDs. The following populations were sorted: hematopoietic stem cells (HSCs, Lineage⁻CD34⁺CD38⁻CD90⁺CD45RA⁻), erythroid progenitors (EPs, Lineage⁻CD34⁺CD38⁺CD41a⁻CD123⁻CD45RA⁻), megakaryocyte progenitors (MkPs, Lineage⁻CD34⁺CD38⁺CD41a⁺) and ProB-cells (Lineage⁻CD34⁺CD19⁺). Representative profiles from one HD.

Percentages indicate the mean frequency ± standard error of the mean (SEM) for all donors used for clonal tracking of lineage contributions, of the gated stem/ progenitor cells relative to total CD34⁺ cells (n = 56 for HSCs, EPs, and MkPs and n = 21 for ProB-cells). **b.** Flow cytometry gating strategy for isolation of mature blood cell lineages from BM MNCs from healthy donors. The following populations were sorted: myeloid-cells (CD14⁺CD33⁺CD3⁻CD19⁻CD56⁻), B-cells (CD19⁺CD3⁻CD33⁻CD56⁻), T-cells (CD3⁺CD8a⁺CD4⁻CD19⁻CD33⁻CD56⁻ and CD3⁺CD4⁺CD8a⁻CD19⁻CD33⁻CD56⁻).

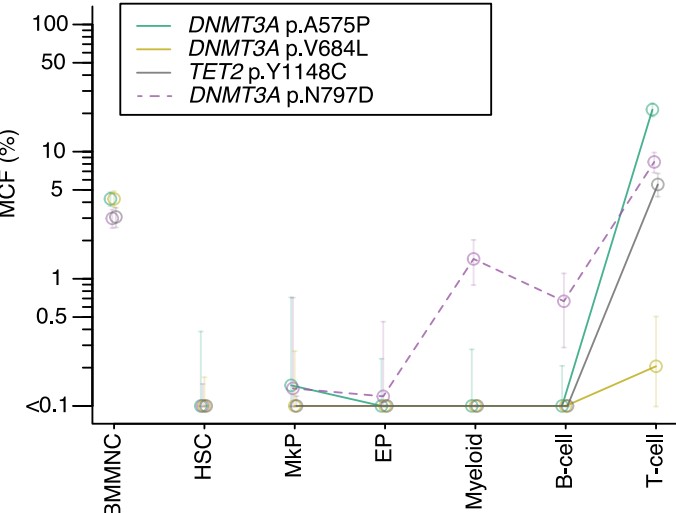

**Extended Data Fig. 4 | Lineage contribution of clones marked by mutations not reliably detected in HSCs.** MCFs (dots) in different blood cell lineages and their 95% credible intervals estimated by Bayes using the number of ddPCR events and number of cells analyzed (error bars) of clones defined by specified CH-driver mutations not confidently detected in Lineage⁻CD34⁺CD38⁻CD90⁺CD45RA⁻ HSCs (n = 4) are shown. Clones observed specifically only in T cells are indicated by solid lines (n = 3).

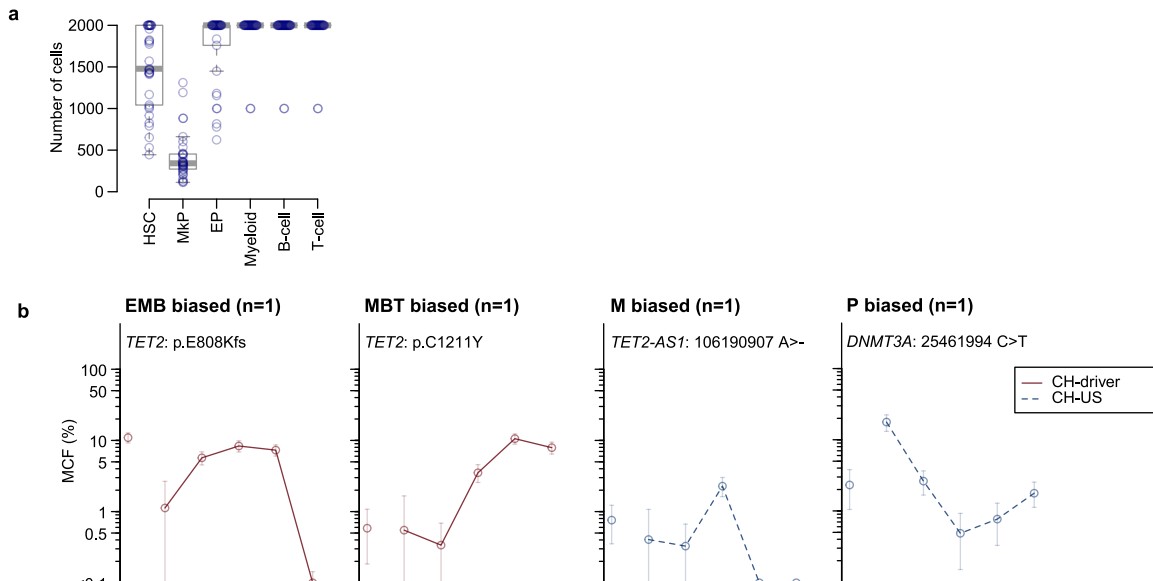

**Extended Data Fig. 5 | Unique lineage contribution patterns of HSC clones. a**. Number of cells subjected to mutational ddPCR analysis for all 33 cases at the first visit analyzed: Lineage⁻CD34⁺CD38⁻CD90⁺CD45RA⁻ HSCs, Lineage⁻CD34⁺CD38⁺CD41⁺ MkP for the platelet (P) lineage, Lineage⁻CD34⁺CD38⁺ CD41⁻CD123⁻CD45RA⁻ EP for the erythroid (E) lineage, CD33⁺CD14⁺ myeloid-cells (M), CD19⁺ B-cells (B) and CD4 or CD8 single positive T-cells (T). The center lines

and boxes denote median values and the first and third quartiles, respectively. Whiskers indicate maximum and minimum values within 1.5× interquartile ranges. **b**. HSC clones with unique lineage replenishment patterns are shown in the same manner as Fig. 2b. Each pattern was observed only for a single HSC clone.

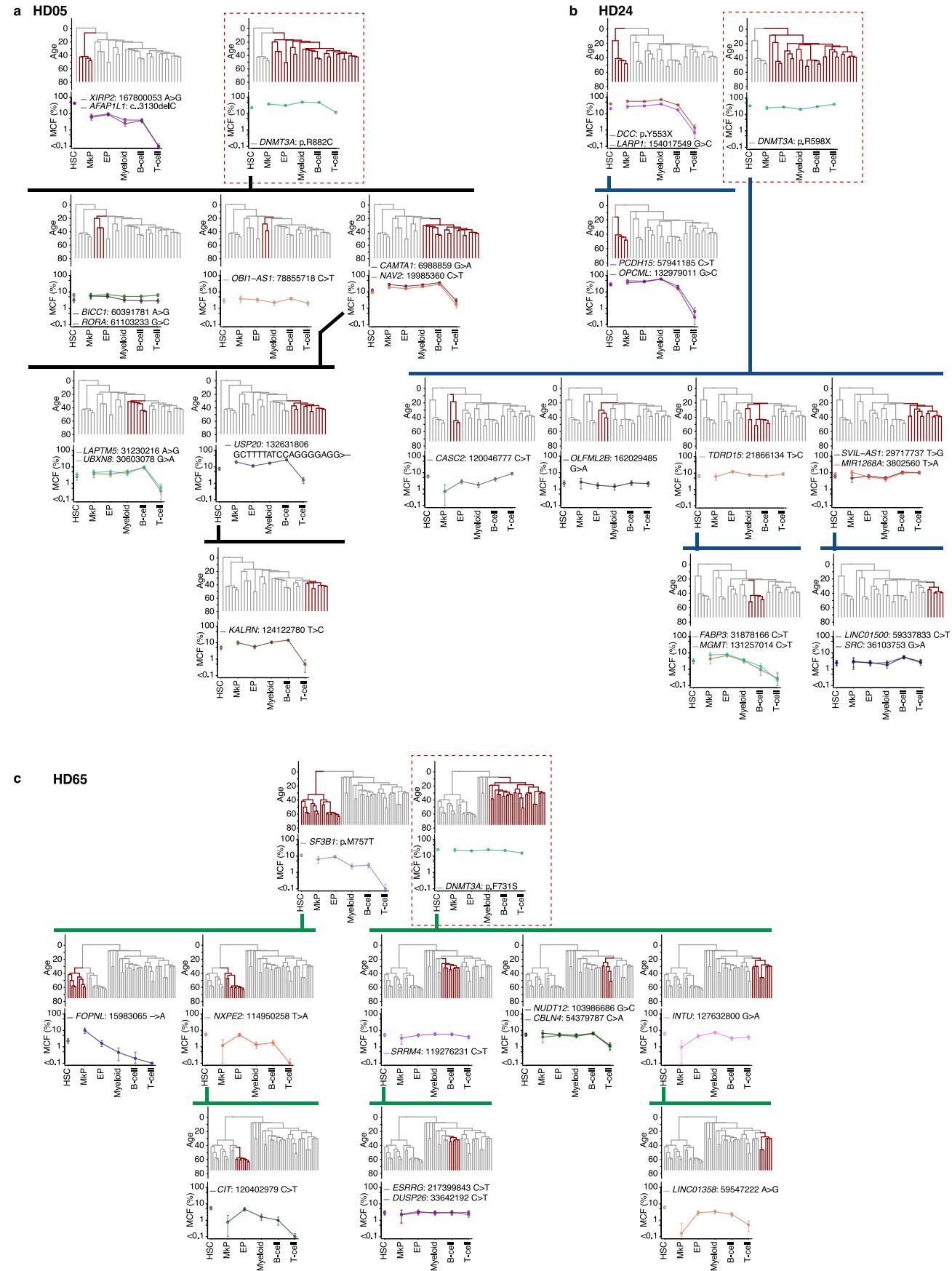

**Extended Data Fig. 6 | See next page for caption.**

**Extended Data Fig. 6 | Phylogenetic history of cases with *DNMT3A*-mutated PEMBT HSC clones.** MCFs in the different blood cell lineages for HSC clones (error bars show 95% credible intervals estimated by Bayes taking into account the number of droplets and the number of input cells (See Methods for details)) and their phylogenetic trees showing the clades where clones (and subclones connected by bold lines) belong (red) from cases with *DNMT3A*-mutated HSC clones showing PEMBT multilineage replenishment; HD05 (**a**), HD24 (**b**),

and HD65 (**c**) from which data are included in Fig. 5c. The panels in which the phylogenetic tree of the ancestor *DNMT3A*-mutated clones are shown are boxed with red dashed lines. The y-axis in the trees is scaled based on age of donor, starting (top of phylogenetic tree) at the time of conception (40 weeks before birth), and ending (bottom of tree) at the time of collection of the investigated BM sample.

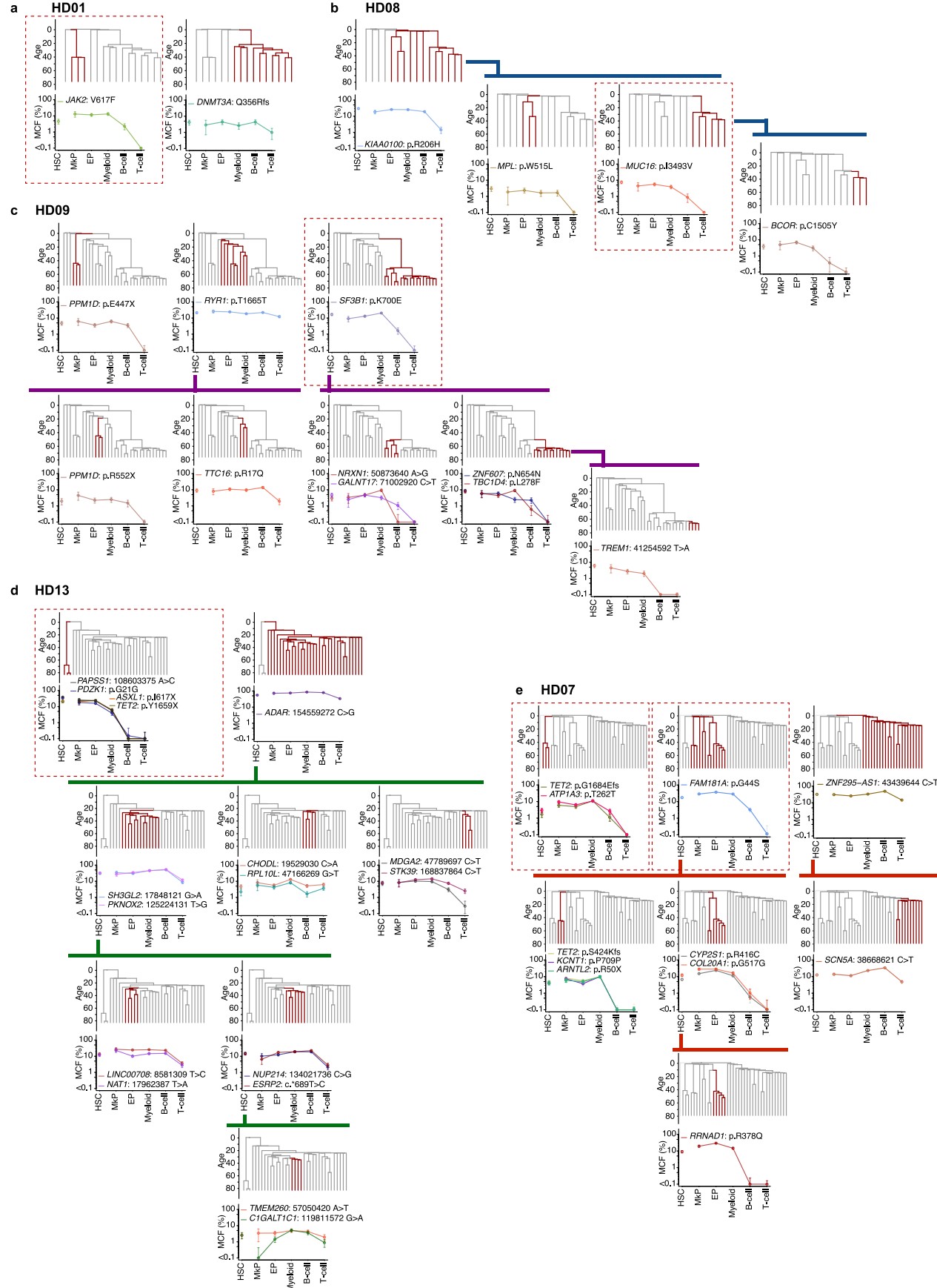

**Extended Data Fig. 7 | See next page for caption.**

**Extended Data Fig. 7 | Phylogenetic history of cases with PEM HSC clones.**
MCFs in the different blood cell lineages for HSC clones with their 95% credible intervals and their phylogenetic trees showing the clades where clones belong (red) from cases with PEM pattern (restricted or biased) clones; HD01 (**a**), HD08 (**b**), HD09 (**c**), HD13 (**d**), and HD07 (**e**), from which data are included in Fig. 5a–b,d. Ancestor PEM clones are boxed with red dashed lines. The y-axis in the trees is scaled based on age of donor, starting (top of phylogenetic tree) at the time of conception (40 weeks before birth), and ending (bottom of tree) at the time of collection of the investigated BM sample.

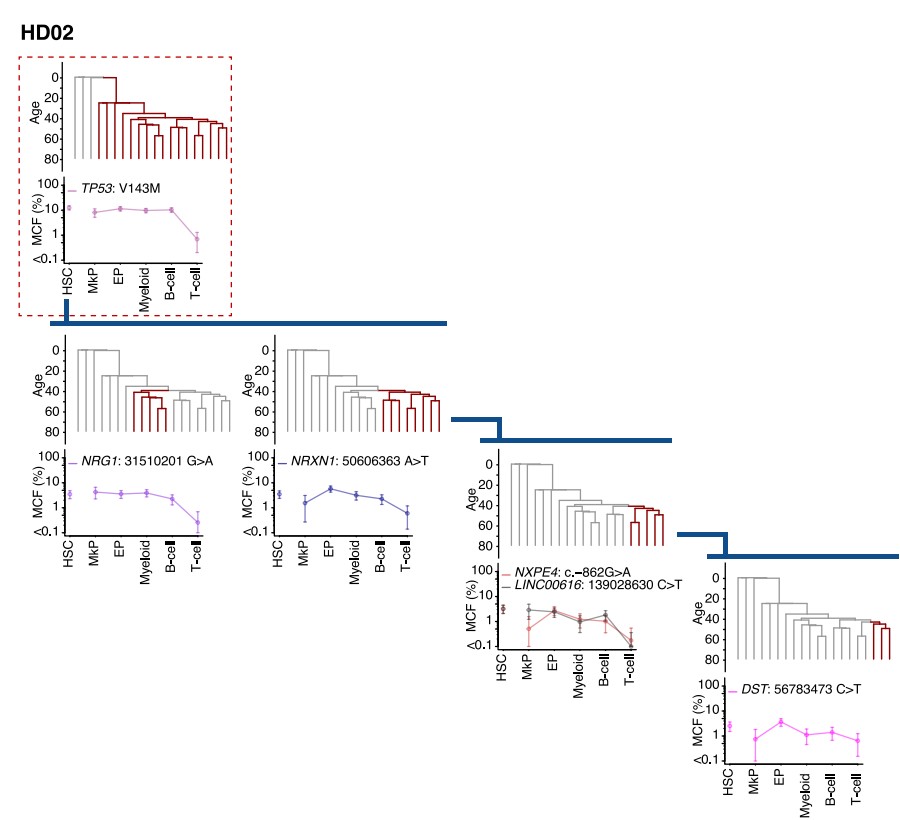

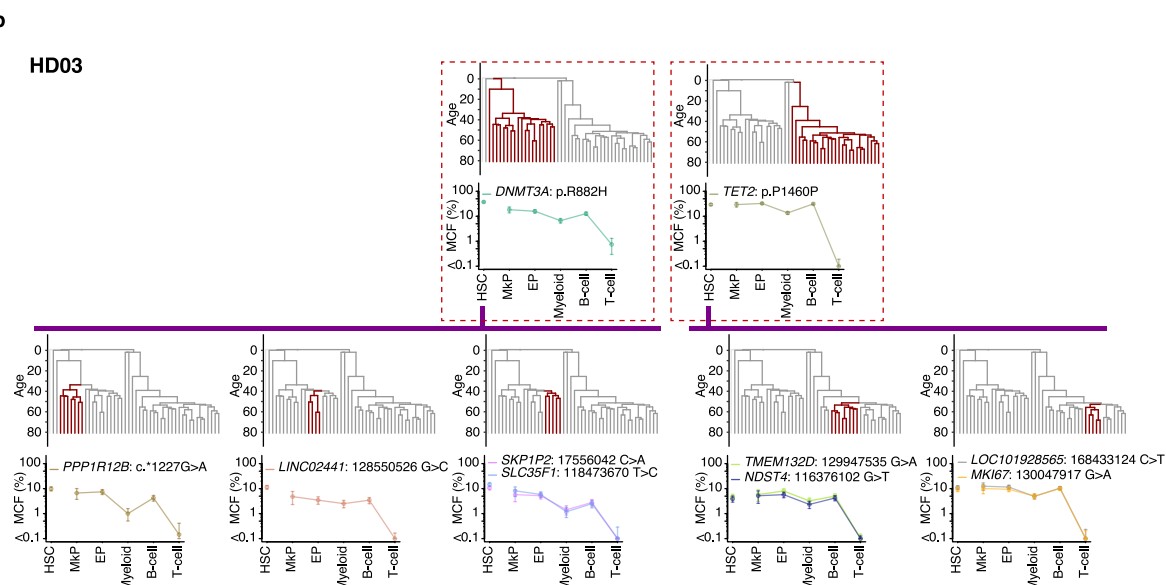

**Extended Data Fig. 8 | Phylogenetic history of PEMB HSC clones.** MCFs in the different blood cell lineages for PEMB-biased/restricted clones with their 95% credible intervals and their phylogenetic trees showing the clades where clones belong (red); HD02 (**a**) and HD03 (**b**). Ancestor PEMB clones are boxed with red dashed lines. The y-axis in the trees is scaled based on age of donor, starting (top of phylogenetic tree) at the time of conception (40 weeks before birth), and ending (bottom of tree) at the time of collection of the investigated BM sample.

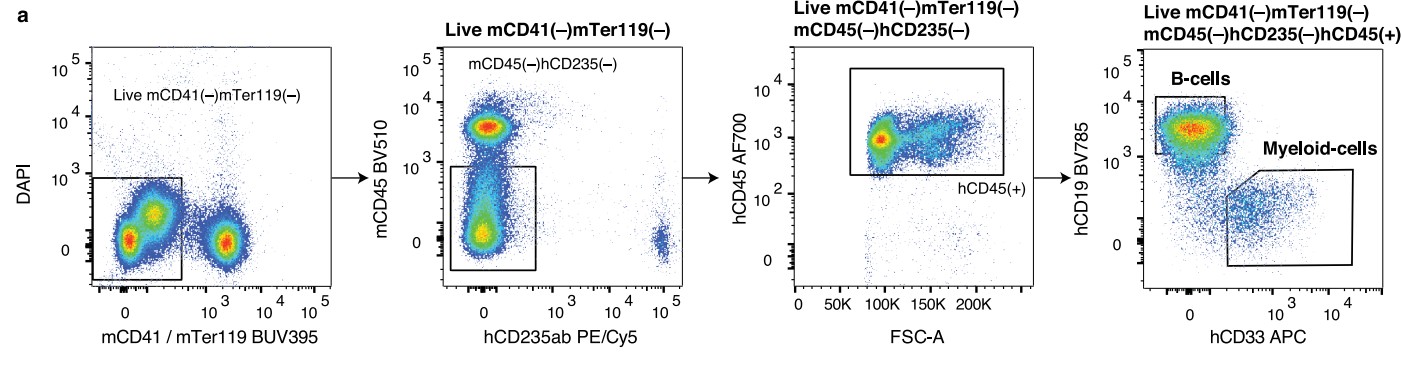

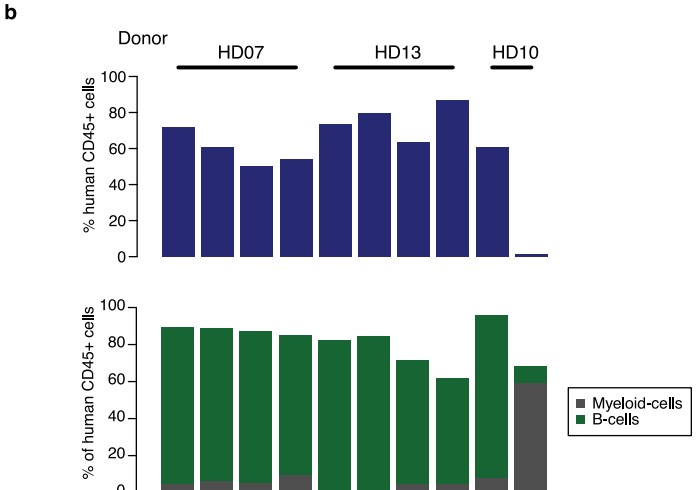

**Extended Data Fig. 9 | Reconstitution of human B lymphopoiesis in NSG mice. a.** Flow cytometry gating strategy for identification and isolation of mature human myeloid (human (h)CD45⁺hCD33⁺hCD19⁻hCD235⁻ mouse (m) Ter119⁻mCD41⁻mCD45⁻) and human B (hCD45⁺hCD19⁺hCD33⁻hCD235⁻ mTer119⁻mCD41⁻mCD45⁻) cells in the BM of NSG mice 10–14 weeks after transplantation of human CD34⁺/CD34⁺CD19⁻ HSPCs. **b.** Percentage of human (h)CD45 reconstitution of total (human+mouse) CD45⁺ cells (upper panel) and distribution of human myeloid and B-cells among human CD45⁺ cells (lower panel) in BM of NSG mice 10–14 weeks after transplantation of human CD34⁺/CD34⁺CD19⁻ hematopoietic stem/progenitor cells from HDs 7, 13 and 10. Each bar represents data from one mouse.

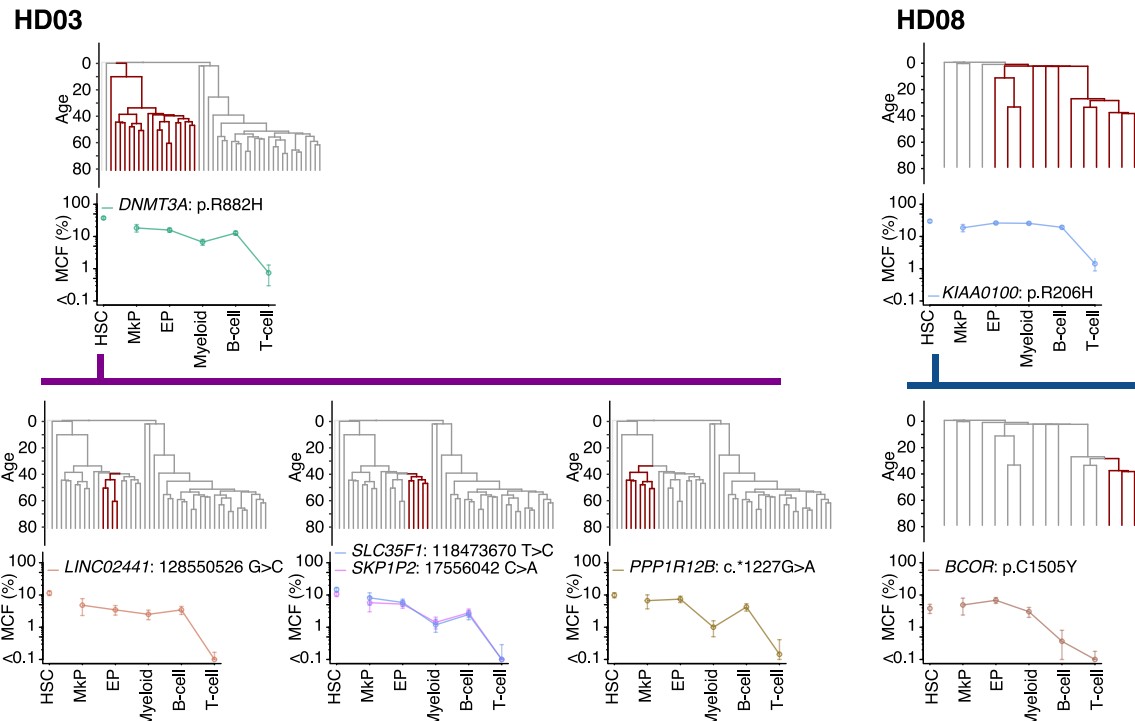

**Extended Data Fig. 10 | Absent T cell contribution from descendant subclones of PEMB-biased ancestral clones.** MCFs with their 95% credible intervals in the different blood cell lineages for PEMB-biased clones from HD03 and HD08 and their descendant clones showing absent contribution to T-cells. Ancestors and their descendant clones are connected by lines of different colors for each donor.

Sten Eirik W. Jacobsen

# Reporting Summary

## Statistics

For all statistical analyses, confirm that the following items are present in the figure legend, table legend, main text, or Methods section.

| n/a | Confirmed | |
|---|---|---|
| ☐ | ☒ | The exact sample size (*n*) for each experimental group/condition, given as a discrete number and unit of measurement |
| ☐ | ☒ | A statement on whether measurements were taken from distinct samples or whether the same sample was measured repeatedly |
| ☐ | ☒ | The statistical test(s) used AND whether they are one- or two-sided *Only common tests should be described solely by name; describe more complex techniques in the Methods section.* |
| ☐ | ☒ | A description of all covariates tested |
| ☐ | ☒ | A description of any assumptions or corrections, such as tests of normality and adjustment for multiple comparisons |
| ☐ | ☒ | A full description of the statistical parameters including central tendency (e.g. means) or other basic estimates (e.g. regression coefficient) AND variation (e.g. standard deviation) or associated estimates of uncertainty (e.g. confidence intervals) |
| ☐ | ☒ | For null hypothesis testing, the test statistic (e.g. *F*, *t*, *r*) with confidence intervals, effect sizes, degrees of freedom and *P* value noted *Give P values as exact values whenever suitable.* |
| ☐ | ☒ | For Bayesian analysis, information on the choice of priors and Markov chain Monte Carlo settings |
| ☒ | ☐ | For hierarchical and complex designs, identification of the appropriate level for tests and full reporting of outcomes |
| ☐ | ☒ | Estimates of effect sizes (e.g. Cohen's *d*, Pearson's *r*), indicating how they were calculated |

*Our web collection on statistics for biologists contains articles on many of the points above.*

## Software and code

Policy information about availability of computer code

| | |
|---|---|
| Data collection | Flow cytometry data: FACSDiva (BD version 8.0.2)<br>DNA sequencing data: NovaSeq 6000 (Illumina)<br>ddPCR data:  QuantaSoft version 1.7.4 and QuantaSoft Analysis Pro version 1.0.596 (Bio-Rad).<br>Genotype data: ddPCR or BioMARK HD (Fluidigm) |
| Data analysis | Flow cytometry data: FACSDiva (version 8.0.2) was analyzed by FlowJo (version 10.10.0)<br>Sequencing reads were mapped to GRCh37 using the Burrows-Wheeler Aligner version: 0.7.17.<br>Error-corrected sequencing: reads with the same unique molecular identifier (UMI) are grouped  Pickard (version 2.20.2) and consensus reads were generated using fgbio (version 0.8.1) . Consensus reads were subjected to indel realignment and base quality score recalibration using GATK3 (version 3.8) and recalculation of MD/NM tags using SAMtools (version 1.9). Mutation calling was performed using EBCall (https://github.com/friend1ws/EBCall). Mutations were annotated using ANNOVAR (version 8 June 2020). After mutation calling, read-based variant allele frequencies were calculated based on the reads using the GenomonMutationFilter version 0.2.8.<br>Common for whole-exome and whole-genome sequencing: PCR duplicates were marked using Biobambam version 2.0.87. Errors associated with enzymatic fragmentation were removed using FADE version 0.2.255. Mutation calling was performed using GenomonFisher (version 0.4.4) (https://github.com/Genomon-Project/GenomonFisher).  Called mutations were annotated using ANNOVAR (version 8 June 2020). After mutation calling, read-based variant allele frequencies were calculated based on the reads using the GenomonMutationFilter version 0.2.8.<br>Specific to whole-exome sequencing data: Copy number analysis was performed using CNACS (https://github.com/OgawaLabTumPath/CNACS).<br>Spedific to whole-genome sequencing data: Copy number analysis was performed using Control-FREEC version 11.632 and ASCAT_R package version 3.1.157. Chromosome Y loss in male donors were evaluated by calculating the sequence depth of chromosomes X and Y using |

SAMtools (version 1.9). Mutational signature analysis was performed using MutationalPatterns version 3.7.0 together with following R packages: ggplot2 (version 3.3.6), biomaRt (version 2.53.3), ccfindR (version 1.17.0), gridExtra (version 2.3), BSgenome.Hsapiens.UCSC.hg19 (version 1.4.3), TxDb.Hsapiens.UCSC.hg19.knownGene (version 3.2.2), BSgenome (version 1.65.2), rtracklayer (version 1.57.0), and NMF (version 0.24.0).

Phylogenetic analysis: Phylogenetic analysis was performed using Sifit (https://github.com/KChen-lab/SiFit). The length of branches were corrected using the "get_corrected_tree" R function downloaded from https://github.com/emily-mitchell/normal_haematopoiesis/tree/main31. Mutations were allocated to branches using the R package "treemut" version 1.1.

ddPCR data: ddPCR data were analyzed using QuantaSoft v1.5.38.1118 software (Bio-Rad). Mutant cell fractions (MCFs) and their credible intervals were generated through Bayesian inference using CmdStan version 2.34.1 based on the number of droplets assigned to the 2D ddPCR quadrants and the number of sorted cells. The Markov Chain Monte Carlo (MCMC) methods implemented in CmdStanR version 0.7.1 were used for parameter estimation.

Statistical analyses were performed using R version 4.2.2. The following open source R packages were used in the analyses presented throughout this paper: magrittr (version 2.0.3), flowCore (version 2.10.0), overlapping (version 2.2), rstan (version 2.32.6), broom (version 1.0.8), bayesplot (version 1.11.1), posterior (version 1.6.1), tidyverse (version 2.0.0), dplyr (version 1.1.4), purr (version 1.0.2), tibble (version 3.2.1), pheatmap (version 1.0.12), stringi (version 1.8.4), stringr (version 1.5.1), ggrepel (version 0.9.5), ggplot2 (version 3.5.1), ggalluvial (version 0.12.5), exactRankTests (version 0.8-35), formattable (version 0.2.1), readxl (version 1.4.3), beeswarm (version 0.4.0), RColorBrewer (version 1.1-3), devtools (version 2.4.5), lmerTest (version 3.1-3), lme4 (version 1.1-35.3), glmmTMB (version 1.1.11), ape (version 5.8 ), ggtree (version 3.6.2), gplots (version 3.2.0), and spdep (version 1.3-3).

For manuscripts utilizing custom algorithms or software that are central to the research but not yet described in published literature, software must be made available to editors and reviewers. We strongly encourage code deposition in a community repository (e.g. GitHub). See the Nature Portfolio guidelines for submitting code & software for further information.

# Data

Policy information about availability of data

All manuscripts must include a data availability statement. This statement should provide the following information, where applicable:

- Accession codes, unique identifiers, or web links for publicly available datasets
- A description of any restrictions on data availability
- For clinical datasets or third party data, please ensure that the statement adheres to our policy

All the DNA sequencing data has been deposited in the Swedish National Data Service (SND) (https://researchdata.se/en, DOI: 10.48723/313d-dd68). Detected mutation list for error-corrected targeted capture sequencing, whole-exome sequencing and single colony whole-genome sequencing, ddPCR result, and codes for data analysis and figure generation are available through SciLifeLab Data Repository (DOI: 10.17044/scilifelab.24745464).

Sequencing data were mapped to combined reference of human genome reference GRCh37 (ftp://ftp.ncbi.nih.gov/genomes/archive/old_genbank/Eukaryotes/vertebrates_mammals/Homo_sapiens/GRCh37/special_requests/GRCh37-lite.fa.gz), Human herpesvirus 4 complete wild type genome (http://www.ncbi.nlm.nih.gov/nuccore/82503188?report=fasta), and the decoy sequence (ftp://ftp.1000genomes.ebi.ac.uk/vol1/ftp/technical/reference/phase2_reference_assembly_sequence/hs37d5cs.fa.gz).

The mutational signature analysis was performed using the combined reference of COSMIC mutational Signatures v2 (https://cancer.sanger.ac.uk/signatures/signatures_v2/) and SBS-blood (https://doi.org:10.1038/s41586-022-05072-7).

736 genes registered in the COSMIC Cancer Gene Census (https://cancer.sanger.ac.uk/census accessed on 2023/03/06) and 95 previously defined myeloid genes (https://doi.org:10.1038/s41586-022-04786-y) were used for the definition of driver mutations.

# Research involving human participants, their data, or biological material

Policy information about studies with human participants or human data. See also policy information about sex, gender (identity/presentation), and sexual orientation and race, ethnicity and racism.

| | |
|---|---|
| Reporting on sex and gender | Information of donor sex is reported in the source data. |
| Reporting on race, ethnicity, or other socially relevant groupings | All the donors were recruited from Sweden and samples were collected at Karolinska Institutet in Sweden. Donor ethnicity is not available for this study. |
| Population characteristics | Information of donor age is reported in the source data. |
| Recruitment | Recruitment of donors was performed under the Stockholm regional ethical review board (EPN 2018/901-31). |
| Ethics oversight | This study was conducted under the Stockholm regional ethical review board (EPN 2018/901-31). |

Note that full information on the approval of the study protocol must also be provided in the manuscript.

# Field-specific reporting

Please select the one below that is the best fit for your research. If you are not sure, read the appropriate sections before making your selection.

☒ Life sciences  ☐ Behavioural & social sciences  ☐ Ecological, evolutionary & environmental sciences

For a reference copy of the document with all sections, see nature.com/documents/nr-reporting-summary-flat.pdf

# Life sciences study design

All studies must disclose on these points even when the disclosure is negative.

| Sample size | No statistical methods were used to determine sample size since this is an exploratory study. We enrolled all the 93 donors recruited from September 2018 to October 2019. |
|---|---|
| Data exclusions | All the enrolled donors were included in this study. <br> We excluded whole-genome sequencing data from 21 colonies which showed low depth or skewed peaks of variant allele frequencies in detected mutations. <br> Colonies derived from the same BM sample from HD03 were sequenced on two separate occasions (several years apart) and mutations shared among colonies sequenced on the later occasion were observed which were absent in the earlier sequencing run. These mutations were therefore presumed to be errors induced through long storage and were consequently excluded. Because these errors could lead to inaccurate age estimation, HD03 was also omitted from the statistical analysis for age dating of HSC clones. |
| Replication | Human CD34+ or CD34+CD19− cells were transplanted into 4 NOD.Cg-Prkdcscidll2rgtm1Wjl/SzJ (NSG) mice per donor as biological replicates. Mutant cell fractions in B cells after transplantation were confirmed by performing ddPCR using biological replicates for sorting. <br> We validated all the mutations detected by error corrected targeted capture sequencing with >2% VAF driver mutations and >1% non-driver mutations by ddPCR. |
| Randomization | Randomization is not relevant to this study since all the donors were healthy and subjected to screening of mutations. |
| Blinding | Blinding is not relevant to this study. |

# Reporting for specific materials, systems and methods

We require information from authors about some types of materials, experimental systems and methods used in many studies. Here, indicate whether each material, system or method listed is relevant to your study. If you are not sure if a list item applies to your research, read the appropriate section before selecting a response.

## Materials & experimental systems

| n/a | Involved in the study |
|---|---|
| ☐ | ☒ Antibodies |
| ☐ | ☒ Eukaryotic cell lines |
| ☒ | ☐ Palaeontology and archaeology |
| ☐ | ☒ Animals and other organisms |
| ☐ | ☒ Clinical data |
| ☒ | ☐ Dual use research of concern |
| ☒ | ☐ Plants |

## Methods

| n/a | Involved in the study |
|---|---|
| ☒ | ☐ ChIP-seq |
| ☐ | ☒ Flow cytometry |
| ☒ | ☐ MRI-based neuroimaging |

## Antibodies

| Antibodies used | Information on the antibodies used in this study, including dilutions, catalog numbers, and lot numbers, is provided in Supplementary Table 5. |
|---|---|
| Validation | All antibodies used in the study were obtained from commercial vendors and were validated by their manufacturers for the application (flow cytometry) and species (mouse) used in this study. Furthermore, all antibodies used have been individually titrated prior to use to identify their optimal concentration in the required application. The specificity of staining was controlled based on simultaneous analysis of cell populations known to lack expression of the relevant antigens. All experiments included fluorescence-minus-one (FMO) controls and, where possible, staining panels included internal controls (known negative and positive populations) to validate specific antibody signals. |

## Eukaryotic cell lines

Policy information about cell lines and Sex and Gender in Research

| Cell line source(s) | Jurkat and K562: both from ATCC |
|---|---|
| Authentication | Authenticated cell lines were purchased from ATCC. |
| Mycoplasma contamination | Cells were regularly tested for mycoplasma contamination and were confirmed negative before experiments. |
| Commonly misidentified lines <br> (See ICLAC register) | Not used in this study. |

# Animals and other research organisms

Policy information about [studies involving animals](studies involving animals); [ARRIVE guidelines](ARRIVE guidelines) recommended for reporting animal research, and [Sex and Gender in Research](Sex and Gender in Research)

| | |
|---|---|
| Laboratory animals | NOD.Cg-PrkdcscidIl2rgtm1Wjl/SzJ (NSG) mice from the Jackson Laboratory |
| Wild animals | No wild animals were used in this study. |
| Reporting on sex | Female mice were used as described in the methods. |
| Field-collected samples | Not relevant for this study. |
| Ethics oversight | All mouse experiments were performed at Karolinska Institutet in Sweden according to the guidelines and obtained permissions from the ethics committees at Stockholms Djurförsöksetiska Nämnd (17978-18 with amendments 18539-21). Mice were maintained in individually ventilated cages with a 12/12 h light/dark cycle, at 22 ± 1 °C and 50% relative humidity. |

Note that full information on the approval of the study protocol must also be provided in the manuscript.

# Clinical data

Policy information about [clinical studies](clinical studies)
All manuscripts should comply with the ICMJE [guidelines for publication of clinical research](guidelines for publication of clinical research) and a completed [CONSORT checklist](CONSORT checklist) must be included with all submissions.

| | |
|---|---|
| Clinical trial registration | Not a clinical trial, and not relevant for the others |
| Study protocol | *Note where the full trial protocol can be accessed OR if not available, explain why.* |
| Data collection | *Describe the settings and locales of data collection, noting the time periods of recruitment and data collection.* |
| Outcomes | *Describe how you pre-defined primary and secondary outcome measures and how you assessed these measures.* |

# Plants

| | |
|---|---|
| Seed stocks | *Report on the source of all seed stocks or other plant material used. If applicable, state the seed stock centre and catalogue number. If plant specimens were collected from the field, describe the collection location, date and sampling procedures.* |
| Novel plant genotypes | *Describe the methods by which all novel plant genotypes were produced. This includes those generated by transgenic approaches, gene editing, chemical/radiation-based mutagenesis and hybridization. For transgenic lines, describe the transformation method, the number of independent lines analyzed and the generation upon which experiments were performed. For gene-edited lines, describe the editor used, the endogenous sequence targeted for editing, the targeting guide RNA sequence (if applicable) and how the editor was applied.* |
| Authentication | *Describe any authentication procedures for each seed stock used or novel genotype generated. Describe any experiments used to assess the effect of a mutation and, where applicable, how potential secondary effects (e.g. second site T-DNA insertions, mosiacism, off-target gene editing) were examined.* |

# Flow Cytometry

## Plots

Confirm that:

☒ The axis labels state the marker and fluorochrome used (e.g. CD4-FITC).

☒ The axis scales are clearly visible. Include numbers along axes only for bottom left plot of group (a 'group' is an analysis of identical markers).

☒ All plots are contour plots with outliers or pseudocolor plots.

☒ A numerical value for number of cells or percentage (with statistics) is provided.

## Methodology

| | |
|---|---|
| Sample preparation | Viably frozen bone marrow mononuclear cells were thawed in a 37°C water bath and washed with Dulbecco's phosphate buffered saline (PBS, Gibco) supplemented with 20% FCS (Sigma-Aldrich) and 100 µg/ml DNase I, Bovine Pancreas (Sigma-Aldrich). For human cells transplanted into NSG mice, BM was isolated 10-14 weeks after transplantation. After 5 minutes incubating in FcR Blocking Reagent, human (Miltenyi Biotec), cells were stained with the fluorescently conjugated monoclonal antibodies for 15 minutes at 4 °C. Stained cells were washed with PBS supplemented with 5% FCS and 2mM EDTA (Invitrogen) and was added with DAPI |

(Invitrogen) for HSPC panel or 7AAD for mature cell panel (Sigma-Aldrich) for identifying live cells just before FACS analysis.

| | |
|---|---|
| Instrument | All flow cytometry experiments were performed on a FACSAria Fusion (BD Biosciences). |
| Software | FACSDiva (version 8.0.2), FlowJo (version 10.10.0), and R package flowCore (version 2.10.0). |
| Cell population abundance | For all the cell sorting, the purities of targeted cell populations were checked prior to cell sorting was done. For populations with ≥50 cells were sorted for purity evaluation, we achieved >97% purity in live cell gating (n=83). |
| | Single cell sorting was performed with index sorting which allows prospective analysis of the cell surface expression of the single cell sorted. Accuracy of single cell deposition in 96 well plates was validated using 488 nm fluorescent beads. |
| Gating strategy | FSC-A/SSC-A was used for gating mononuclear cells. Doublets were excluded. 7AAD or DAP positive cells were gated out to exclude non-viable cells. |

Live cells were gated according to the following expression markers.

Haematopoietic stem cells (HSCs): Lineage–CD34+CD38–CD90+CD45RA–
Megakaryocyte progenitor cells (MkP): Lineage–CD34+CD38+CD41a+
Erythroid progenitor cells (EP): Lineage–CD34+CD38+CD41a–CD123–CD45RA–
ProB cells (ProB): Lineage–CD34+CD19+
Myeloid cells: CD14+CD33+CD3–CD19–CD56–
B cells: CD19+CD3–CD33–CD56–
T cells: CD3+CD8a+CD4–CD19–CD33–CD56– and CD3+CD4+CD8a–CD19–CD33–CD56–
Lineage markers include CD2, CD3, CD4, CD7, CD8a, CD10, CD11b, CD14, CD19, CD20, CD56, and CD235a.
Isolation of humane cells after transplantation into NSG mice: Human myeloid cells (mTer119–mCD41–mCD45–hCD235ab–hCD45+CD33+CD19–CD36–) and human B cells (mTer119–mCD41–mCD45–hCD235ab–hCD45+CD33–CD19+CD36–)

☒ Tick this box to confirm that a figure exemplifying the gating strategy is provided in the Supplementary Information.

