## [Peer Review File · Nature Genetics]

Stable clonal contribution of lineage-restricted stem cells to human hematopoiesis

Corresponding Author: Professor Sten Eirik Jacobsen

This manuscript has been previously reviewed at another journal. This document only contains information relating to versions considered at Nature Genetics.

Version 0:

Decision Letter:

29th Jul 2025

Dear Professor Jacobsen,

Your Article, "Stable clonal contribution of lineage-restricted stem cells to steady-state human hematopoiesis" has now been seen by 2 referees. You will see from their comments below that while they find your work of interest, some important points are raised. We are interested in the possibility of publishing your study in Nature Genetics, but would like to consider your response to these concerns in the form of a revised manuscript before we make a final decision on publication.

To guide the scope of the revisions, the editors discuss the referee reports in detail within the team, including with the chief editor, with a view to identifying key priorities that should be addressed in revision and sometimes overruling referee requests that are deemed beyond the scope of the current study. In this case, we ask you to address the remaining requests by Reviewers in full. We hope that you will find the prioritized set of referee points to be useful when revising your study. Please do not hesitate to get in touch if you would like to discuss these issues further.

We therefore invite you to revise your manuscript taking into account all reviewer and editor comments. Please highlight all changes in the manuscript text file. At this stage we will need you to upload a copy of the manuscript in MS Word .docx or similar editable format.

*2) If you have not done so already please begin to revise your manuscript so that it conforms to our Article format instructions, available

[here](http://www.nature.com/ng/authors/article_types/index.html).

*3) Include a revised version of any required Reporting Summary: <https://www.nature.com/documents/nr-reporting-summary.pdf>

EXTENDED DATA FIGURES

Link Redacted

We hope to receive your revised manuscript within four to eight weeks. If you cannot send it within this time, please let us know.

Sincerely,
Chiara

Chiara Anania, PhD
Associate Editor
Nature Genetics
<https://orcid.org/0000-0003-1549-4157>

Reviewers' Comments:

Reviewer #1 (Remarks to the Author):

This manuscript by Yoshizato et al. presents new evidence of fate-restricted stem cell clones in the context of native human hematopoiesis. Critically, it is the first study to include extensive analyses of the Platelet and Erythroid lineages. I have reviewed this manuscript before and I still consider it constitutes one of the most robust demonstrations to date that fate-biased stem cells exist and contribute to human hematopoiesis in real individuals. The biggest caveat remained that clones are being traced in aged individuals and therefore, the existence of these fate-biased clones in healthy young hematopoiesis is still uncertain. However, the researchers have put substantial new data and analysis that strengthen the universality of their conclusions. Namely, they have presented evidence of a few cases where different biased sub-clones could be traced back to a shared common ancestor, while other siblings of such ancestor did not display a fate-biased behavior. In practice, this demonstrates that multiple daughter HSCs retained the fate bias of their parental cell (it would be helpful to have a proper quantification of this for the various cell-type MCFs across a mutationally-defined tree using Moran's I or similar auto-correlation analysis). In any case, there are two possible explanations to the phenomenon: either a) that the ancestral HSC was already fate-biased when that individual was young, or b) the loss of lymphoid output was pre-determined within that ancestral HSC, such that all the HSC daughters lost it roughly in the same way. It is impossible to distinguish between the two with present data and methods. However, researchers also presented new evidence that might solve this conundrum: they show that fate-biased clones (PEMB, PEM) typically emerge a decade or so after the PEMBT clones. This could be important to show while controlling for the mutated gene (i.e. only for Dnmt3a mutant clones, or only for CH-US clones), but still, it suggests that fate-restriction occurs to clones as they expand through time: some clones expand yet retain multilineage capacity... some clones expand but lose certain fates. This could hint at some time-controlled processes as the mechanism of human platelet/myeloid HSC fate-bias, decades before aging is observed, with big implications for understanding and manipulating these HSCs to modulate human healthspan.

In sum, I remain convinced that this is an outstanding contribution that provides very important pieces of the puzzle and it is carefully done with state-of-the-art techniques across a sufficiently robust number of clones and individuals.

Reviewer #2 (Remarks to the Author):

This revised manuscript presents a technically sophisticated and timely study of clonal lineage behavior in human hematopoiesis. By leveraging somatic mutation-based barcoding, single-cell whole-genome sequencing, and serial sampling over multiple years, the authors track the lineage output of individual HSC clones in aged individuals. They identify recurrent clonal patterns, balanced (PEMBT), partially restricted (PEMB), and myeloid-restricted (PEM), and argue that these patterns are intrinsically programmed and temporally stable. The integration of prospective and retrospective lineage tracing is impressive.

That said, several key issues remain unresolved and deserve attention:

- Framing and Scope: The manuscript aims to elucidate principles of steady-state human hematopoiesis but draws exclusively from aged individuals, most with CH-driver mutations. While the revised version acknowledges this more clearly, the extrapolation to native hematopoiesis across the lifespan remains overconfident. The study is best understood as a rigorous investigation of aged hematopoiesis in CH-positive individuals, and the text should consistently reflect this.
- Driver Mutations and Lineage Fate: The authors argue that lineage restriction patterns are not driven by CH mutations, citing overlap between PEM/PEMB patterns and both driver-positive and CH-US clones. However, this conclusion oversimplifies a more complex and well-documented reality. Prior work has demonstrated gene- and variant-specific effects on lineage bias. TET2 and ASXL1 mutations often exhibit myeloid bias (Jakobsen et al., Cell Stem Cell 2024; Buscarlet et al., Blood 2018), JAK2V617F is strongly myeloid-restricted, DNMT3A R882 variants have shown skewing toward myeloid/megakaryocytic output (Nam et al., Nat Genet 2022). Crucially, the current manuscript itself contains a counterexample: a PEMB-pattern R882 clone in HD03, which the authors' own phylogeny dates to early life (Extended Data Fig. 7). Additionally, in Sup. Fig. 1b, HD24, it is shown that the clones DCC and LARP1 (CH-US) display a PEMB reconstitution pattern even though their acquisition has been estimated very early in life. The same (early acquisition and lineage biased reconstitution profile) is shown by a JAK2V617F mutation in Sup. Fig.2a, which is another known myeloid-biased driver. This directly contradicts the proposed model that early acquisition predicts balanced PEMBT output. Rather than pooling all mutations together, a variant-level analysis is warranted, or at minimum a qualified discussion of exceptions to their rule.
- Inflammation and the Aged Microenvironment: The role of extrinsic factors such as inflammation is not addressed, yet it is important to clonal behavior in aged hematopoiesis. There is robust evidence linking CH to inflammatory signaling and downstream lineage skewing (see Avagyan & Zon, Trends Cell Biol 2023; Jaiswal & Libby, Nat Rev Cardiol 2019). Given that this study draws entirely from elderly individuals, where systemic inflammation is common, the possibility that lineage fate is shaped or stabilized by microenvironmental context should be acknowledged. Even if cell-intrinsic features contribute to stability, they may not be acting in isolation. A brief discussion would help ground the "intrinsic programming" claim in a more complete biological framework.

Other issues:

- Terminology: The distinction between "lineage restriction" (no output) and "bias" (>5-fold skew) is formally defined, but inconsistently applied across the manuscript. For example, PEMB clones are described both as having "no T cells" (suggesting restriction) and as biased (Fig. 2a vs. 2b). This contributes to conflicting clone counts and classification ambiguities. This inconsistency weakens interpretability, especially given that clone classification underpins the study's central conclusions.
- The dataset includes a preponderance of female donors. Recent work (Furer et al., Nat Med 2025) suggests sex-specific differences in clonal behavior, including more pronounced myeloid bias in males during aging. The authors should briefly comment on whether the observed patterns could be influenced by sex distribution.
- Fig.2e: The authors conclude that PEMB and PEM clones are not a result of the acquisition of CH drivers. From a different perspective though, PEMBT or balanced clones are marked solely by CH-US or DNMT3A mutations. Therefore, the presence of any other CH driver in the clones potentially causes lineage biases. Additionally, DNMT3A mutations are observed across all groups (PEMBT-PEMB-PEM), which is consistent with literature stating that DNMT3A mutations might not be causing any lineage biases to the carrying clones (Jakobsen et al., Cell Stem Cell; Buscarlet et al., Blood, 2018), or (for specific variants such as R882) myeloid/megakaryocytic bias (Nam et al., Nat. Genet, 2022). Therefore, at the time being, the argument that CH drivers do not cause any skewing in the lineage production of the carrying clones is an overinterpretation and, although it might be indeed the case, it should be further supported by data and analysis.
- Page 14, last sentence "The stability of HSC clones over time was not restricted to lineage replenishment patterns, but also with regard to clonal size." We do know that the clones (especially the ones carrying known drivers) will expand (Fabre et al., Nature, 2022; Robertson et al., Nat Med). How do the authors explain this observation?
- Page 73, legend Extended Data Figure 2d: "Targeted genes are indicated by different colors". That doesn't seem to be the case. Different colors mark mutation types.
- Typo in Discussion, page 15: "Thus, by the time that a clone replenished from a single human HSC has become traceable through the mutational lineage fate mapping one is tracing the lineage contribution of an expanded clonal family of HSCs derived from a single HSC rather than a single HSC."

Version 1:

Decision Letter:

Our ref: NG-A69292R

28th Aug 2025

Dear Dr. Jacobsen,

Thank you for submitting your revised manuscript "Stable clonal contribution of lineage-restricted stem cells to human hematopoiesis" (NG-A69292R). We found that the paper has improved in revision, and therefore we'll be happy in principle to publish it in Nature Genetics, pending minor revisions to comply with our editorial and formatting guidelines.

Thank you again for your interest in Nature Genetics. Please do not hesitate to contact me if you have any questions.

Congratulations!

Sincerely,
Chiara

Chiara Anania, PhD
Associate Editor
Nature Genetics
<https://orcid.org/0000-0003-1549-4157>

RESPONSES TO REVIEWERS. All revisions have been underlined in the revised manuscript.

Referee #1 (Alejo E. Rodriguez Fraticelli) (Remarks to the Author):

This manuscript by Yoshizato et al. presents new evidence of fate-restricted stem cell clones in the context of native human hematopoiesis. Critically, it is the first study to include extensive analyses of the Platelet and Erythroid lineages. I have reviewed this manuscript before and I still consider it constitutes one of the most robust demonstrations to date that fate-biased stem cells exist and contribute to human hematopoiesis in real individuals. The biggest caveat remained that clones are being traced in aged individuals and therefore, the existence of these fate-biased clones in healthy young hematopoiesis is still uncertain. However, the researchers have put substantial new data and analysis that strengthen the universality of their conclusions. Namely, they have presented evidence of a few cases where different biased sub-clones could be traced back to a shared common ancestor, while other siblings of such ancestor did not display a fate-biased behavior. In practice, this demonstrates that multiple daughter HSCs retained the fate bias of their parental cell (it would be helpful to have a proper quantification of this for the various cell-type MCFs across a mutationally-defined tree using Moran's I or similar auto-correlation analysis). In any case, there are two possible explanations to the phenomenon: either a) that the ancestral HSC was already fate-biased when that individual was young, or b) the loss of lymphoid output was pre-determined within that ancestral HSC, such that all the HSC daughters lost it roughly in the same way. It is impossible to distinguish between the two with present data and methods. However, researchers also presented new evidence that might solve this conundrum: they show that fate-biased clones (PEMB, PEM) typically emerge a decade or so after the PEMBT clones. This could be important to show while controlling for the mutated gene (i.e. only for Dnmt3a mutant clones, or only for CH-US clones), but still, it suggests that fate-restriction occurs to clones as they expand through time: some clones expand yet retain multilineage capacity... some clones expand but lose certain fates. This could hint at some time-controlled processes as the mechanism of human platelet/myeloid HSC fate-bias, decades before aging is observed, with big implications for understanding and manipulating these HSCs to modulate human healthspan.

In sum, I remain convinced that this is an outstanding contribution that provides very important pieces of the puzzle and it is carefully done with state-of-the-art techniques across a sufficiently robust number of clones and individuals.

RESPONSE:

We appreciate that this reviewer rated the revised version of our manuscript even more highly than the original submission and highlighted that he thinks the revised manuscript “*constitutes one of the most robust demonstrations to date that fate-biased stem cells exist and contribute to human hematopoiesis in real individuals*” and represents “*an outstanding contribution that provides very important pieces of the puzzle and it is carefully done with state-of-the-art techniques across a sufficiently robust number of clones and individuals*”.

One important point brought up by this reviewer is that, although he agrees that we with the retrospective phylogenetic analysis have provided compelling evidence that lineage fate restriction can be programmed at an early age (by demonstrating that multiple daughter hematopoietic stem cells (HSCs) retain the fate bias of their parental HSC), there are two alternative explanations to this observation. One is that the ancestral HSC was already fate-biased when that individual was young; the other is that the loss of lymphoid (B and/or T cell) output was pre-determined within that ancestral HSC, but that all the HSC daughters lost it later albeit roughly in the same way and timeframe. While, as he emphasizes, it is impossible to distinguish between these two possibilities with existing methodology, we now emphasize this significant point in the revised discussion (page 16).

Related to this, as already discussed in the manuscript (page 17), prospective studies in mice have demonstrated the presence of lineage-restricted clones even at young ages (Carrelha et al. Nature 2018 554(7690) 106-111), and notably, the patterns of lineage restriction observed in mice closely resemble the human patterns revealed in our study. Collectively, these findings support that lineage restriction is not a phenomenon unique to aging.

Following this reviewer’s insightful suggestion, we have also conducted a spatial autocorrelation analysis of phylogenetic contribution using Moran’s I test. Because the number of analyzed clades per case was relatively small, we combined all cases into a single analysis. In doing so, we set the weights between clades from different cases to zero and evaluated the correlation between mutant cell fractions (MCFs) of each lineage (normalized to HSCs) and phylogenetic distances measured in terms of age (please see the revised Methods section for details). The results of Moran’s I test are stated in the revised manuscript (page 13), summarized in the table below and have been incorporated as a new **Supplementary Table 3** in the revised manuscript. Importantly, all tests demonstrated strong and statistically significant correlations.

Lineage	Moran I statistic	P value	Adjusted P value
MkP	0.157	0.024	0.025
EP	0.156	0.025	0.025
Myeloid cell	0.494	1.42×10^{-8}	3.56×10^{-8}
B cell	0.601	5.08×10^{-12}	2.54×10^{-11}
T cell	0.478	3.01×10^{-8}	5.02×10^{-8}

P values were adjusted by Benjamini-Hochberg (BH) procedure
MkP: megakaryocyte progenitor, EP: erythroid progenitor

Referee #2 (Original Referee #3) (Remarks to the Author)

This revised manuscript presents a technically sophisticated and timely study of clonal lineage behavior in human hematopoiesis. By leveraging somatic mutation–based barcoding, single-cell whole-genome sequencing, and serial sampling over multiple years, the authors track the lineage output of individual HSC clones in aged individuals. They identify recurrent clonal patterns, balanced (PEMBT), partially restricted (PEMB), and myeloid-restricted (PEM), and argue that these patterns are intrinsically programmed and temporally stable. The integration of prospective and retrospective lineage tracing is impressive. That said, several key issues remain unresolved and deserve attention:

GENERAL RESPONSE (revisions have been underlined in the revised manuscript):

We are grateful for this reviewer’s positive evaluation of the technical aspects and conceptual advances in the revised manuscript. We also value this reviewer’s comments regarding the remaining issues that should be addressed. We have carefully revised the manuscript to respond to each of the points as outlined in our point-by-point responses below.

Main points:

1) Framing and Scope: The manuscript aims to elucidate principles of steady-state human hematopoiesis but draws exclusively from aged individuals, most with CH-driver mutations. While the revised version acknowledges this more clearly, the extrapolation to native hematopoiesis across the lifespan remains overconfident. The study is best understood as a rigorous investigation of aged hematopoiesis in CH-positive individuals, and the text should consistently reflect this.

RESPONSE: While we appreciate that the reviewer finds that we had already more clearly acknowledged this limitation in the revised manuscript, we have now further clarified this in the new revised manuscript, in part by ensuring that we have consistently clarified throughout the Results section that our studies were limited to studies of aged individuals, and in part in the revised Discussion in which we now further emphasize that while the extended phylogenetic analysis allows retrospective inferences regarding clonal lineage restrictions and contributions also at young age (as emphasized by reviewer 1), these inferences have limitations which we now highlight more specifically (revised text, page 17). We have also slightly modified the title of the article by removing the word “steady-state” since this word might give the impression that we have primarily studied hematopoietic stem cells (HSCs) without clonal hematopoiesis (CH) mutations.

2) Driver Mutations and Lineage Fate: The authors argue that lineage restriction patterns are not driven by CH mutations, citing overlap between PEM/PEMB patterns and both driver-positive and CH-US clones. However, this conclusion oversimplifies a more complex and well-documented reality. Prior work has demonstrated gene- and variant-specific effects on lineage bias. TET2 and ASXL1 mutations often exhibit myeloid bias (Jakobsen et al., Cell Stem Cell 2024; Buscarlet et al., Blood 2018), JAK2V617F is strongly myeloid-restricted, DNMT3A R882 variants have shown skewing toward myeloid/megakaryocytic output (Nam et al., Nat Genet 2022). Crucially, the current manuscript itself contains a counterexample: a PEMB-pattern R882 clone in HD03, which the authors’ own phylogeny dates to early life (Extended Data Fig. 7). Additionally, in Sup. Fig. 1b, HD24, it is shown that the clones DCC and LARP1 (CH-US) display a PEMB reconstitution pattern even though their acquisition has been estimated very early in life. The same (early acquisition and lineage biased reconstitution profile) is shown by a JAK2V617F mutation in Sup. Fig.2a, which is another known myeloid-biased driver. This directly contradicts the proposed model that early acquisition predicts balanced PEMBT output. Rather than pooling all mutations together, a variant-level analysis is warranted, or at minimum a qualified discussion of exceptions to their rule.

RESPONSE:

We agree that we could have included a more balanced discussion of the potential impact of different CH/driver mutations on the observed lineage restrictions/biases. As this reviewer pointed out, our investigations cannot exclude the effect of CH-driver (nor CH of undetermined significance (CH-US)) mutations on the observed lineage biases, and indeed as the reviewer points out several CH mutations have been shown to have effects on lineage development and as such might contribute to the lineage biases observed. This is something we more clearly emphasize in the revised manuscript (pages 8 and 17), in which we also cite some of the relevant references highlighted by the reviewer. However, the effects that have been reported of CH mutations on lineage development (including those cited by the reviewer) have been much less prominent than the lineage-restrictions and strong biases observed in our manuscript. Moreover, the observation that clones marked with CH-US are present across all major lineage restriction and bias patterns (PEMBT, PEMB, and PEM) and the absence of significant enrichment for specific CH mutations for the PEMB and PEM patterns suggests that the observed lineage restrictions or major biases are largely acquired independently of CH-driver mutations. Never-the-less we agree and point out in the revised manuscript that the exact effects of different CH (and other) driver mutations on HSC lineage replenishment remains to be resolved, (page 17).

Regarding the suggestion to evaluate the potential effects of distinct driver mutations, we agree that this would be of interest, but with exception of *DNMT3A* for which such analysis is shown (**Fig. 3a–b**), the number of clones with each mutation was too small, to make assessment at the individual variant level meaningful. Considering that, we are pleased that the reviewer indicated that a more balanced discussion would be acceptable.

3) Inflammation and the Aged Microenvironment: The role of extrinsic factors such as inflammation is not addressed, yet it is important to clonal behavior in aged hematopoiesis. There is robust evidence linking CH to inflammatory signaling and downstream lineage skewing (see Avagyan & Zon, Trends Cell Biol 2023; Jaiswal & Libby, Nat Rev Cardiol 2019). Given that this study draws entirely from elderly individuals, where systemic inflammation is common, the possibility that lineage fate is shaped or stabilized by microenvironmental context should be acknowledged. Even if cell-intrinsic features contribute to stability, they may not be acting in isolation. A brief discussion would help ground the “intrinsic programming” claim in a more complete biological framework.

RESPONSE:

While several findings reported in our manuscript (not the least the transplantation experiments; **Fig. 4c–e**, and text pages 10–11) as well as studies of lineage bias in mice (Carrelha et al. Nature 2018 554(7690) 106–111) strongly support that the observed HSC lineage restriction and biases are largely intrinsically programmed, we fully agree with the reviewer that also extrinsic factors are likely to play an important role. Although not assessed in our studies, there is compelling evidence in the literature that extrinsic factors, including inflammation and the microenvironment, are likely to impact not only on HSC clonal expansion but also on lineage biases, something we now reference and highlight in the discussions (text page 17).

Minor:

4) Terminology: The distinction between “lineage restriction” (no output) and “bias” (>5-fold skew) is formally defined, but inconsistently applied across the manuscript. For example, PEMB clones are described both as having “no T cells” (suggesting restriction) and as biased (Fig. 2a vs. 2b). This contributes to conflicting clone counts and classification ambiguities. This inconsistency weakens interpretability, especially given that clone classification underpins the study’s central conclusions.

RESPONSE:

Fig. 2a only summarizes the initial lineage analysis for all investigated clones, as the presence and absence of each of the 5 cell lineages, prior to assessing the relative contribution of each clone to the

different lineages as defined and described in **Fig. 2b**. In **Fig. 2b** we also distinguish between “lineage-restricted” clones that entirely lack detectable contribution to one or more lineages (B and T cells) or “lineage-biased” clones that are heavily biased with minimal contribution to one or more lineages. While we think it is noteworthy to emphasize that some clones are fully lineage-restricted, showing no contribution to T cells or T and B cells, the lineage-bias as defined is also extensive, and therefore not surprisingly we (only) observe exactly the same (PEM and PEMB) patterns of lineage-restriction and lineage-bias (as stated on page 7). Moreover, and also not surprising, our phylogenetic analysis demonstrates that lineage-biased clones with time can become fully lineage-restricted (**Fig. 5**, pages 12–13). Therefore, for certain of the analysis in the manuscript, not the least for statistical purposes we have pooled clones that have the same lineage bias and restriction. We do however agree that it is important that we make it clear where this has been done, so we have throughout revised the text and figure legends where this is relevant.

5) The dataset includes a preponderance of female donors. Recent work (Furer et al., Nat Med 2025) suggests sex-specific differences in clonal behavior, including more pronounced myeloid bias in males during aging. The authors should briefly comment on whether the observed patterns could be influenced by sex distribution.

RESPONSE:

It is true that our studies included predominantly female donors, and this combined with the heterogeneity of CH mutations, their timing and other variables precluded an analysis of potential sex-specific differences. Since, as the reviewer points out, other recent studies (Furer et al. Nat Med 2025 31(7) 2442-2451; Stomper et al. Cell Rep 2025 44(4) 115494) have implicated sex specific differences in lineage biases, we have added a point in the revised discussion (page 17) where we reference these studies, and emphasize that there might be sex differences in lineage bias/restriction patterns which our study was not able to assess because of the female dominance among the donors investigated.

6) Fig.2e: The authors conclude that PEMB and PEM clones are not a result of the acquisition of CH drivers. From a different perspective though, PEMBT or balanced clones are marked solely by CH-US or DNMT3A mutations. Therefore, the presence of any other CH driver in the clones potentially causes lineage biases. Additionally, DNMT3A mutations are observed across all groups (PEMBT-PEMB-PEM), which is consistent with literature stating that DNMT3A mutations might not be causing any lineage biases to the carrying clones (Jakobsen et al., Cell Stem Cell; Buscarlet et al., Blood, 2018), or (for specific variants such as R882) myeloid/megakaryocytic bias (Nam et al., Nat. Genet, 2022). Therefore, at the time being, the argument that CH drivers do not cause any skewing in the lineage production of the carrying clones is an overinterpretation and, although it might be indeed the case, it should be further supported by data and analysis.

RESPONSE:

We agree that further studies are needed to determine the impact of specific CH mutations on lineage bias, and we refer to our response to Major point 2, where we address this in detail. As we emphasize there and the in the further revised manuscript (page 17), we do not intend to claim that CH-drivers are not impacting on lineage skewing, on the contrary previous studies suggest that they might have an impact (Challen et al. Nat Genet 2011 44(1) 23-31; Fujino et al. Nat Commun 2021 12(1) 1826; Moran-Crusio et al. Cancer Cell 2011 20(1) 11-24; Quivoron et al. Cancer Cell 2011 20(1) 25-38). Rather, we are asserting that our findings strongly support that the observed major lineage restrictions/biased are unlikely to have been primarily caused by specific mutated genes and also that the mutational acquisition age plays a significant role for the observed lineage replenishment patterns (**Fig. 3**, text pages 9–10). The observation of the same HSC lineage restriction/bias patterns in young adult mice (Carrelha et al. Nature

2018 554(7690) 106-111), where CH-drivers are rare, provides further evidence for the role of intrinsic (and extrinsic) determinants (yet to be determined) other than CH-driver mutations. However, as stated above, this does not mean that we are denying that these mutations might also influence lineage biases, something we now emphasize more clearly in the further revised discussion (page 17).

7) Page 14, last sentence “The stability of HSC clones over time was not restricted to lineage replenishment patterns, but also with regard to clonal size.” We do know that the clones (especially the ones carrying known drivers) will expand (Fabre et al., Nature, 2022; Robertson et al., Nat Med). How do the authors explain this observation?

RESPONSE:

This is an important point, as it might come as a surprise that we find that also the clonal size is stable over the (up to) 5 years analysis in our manuscript. We therefore now address this point in the revised manuscript (page 14). While it is true that CH mutations in general give a clonal advantage, this is in most cases rather limited in the short perspective, and in fact this helps to explain why CH is almost exclusively observed in elderly individuals despite the corresponding CH mutations typically being targeted to a HSC several decades earlier (Fabre et al. Nature 2022 606(7913) 335-342; Robertson et al. Nat Med 2022 28(7) 1439-1446; Watson et al. Science 2020 367(6485) 1449-1454) (**Fig. 3a**). Fabre et al., Nature, 2022 reported that *DNMT3A* and *TP53* mutated clones expand at a rate of approximately 5% more than non-mutated clones per year, and other CH mutations including *TET2*, *ASXL1*, *PPM1D*, and *SF3B1* at about 10% annually. Over the maximum 5-year observation period of our serial analysis, the expected expansion should therefore be 28% (at 5%) and 61% (at 10%), if competing only with non-mutated clones. In light of limited cell numbers being evaluated and likely technical variation, the absence of noticeable changes over 5 years appears reasonable. Additionally, as hematopoiesis in aged individuals is almost invariably oligoclonal (Mitchell et al. Nature 2022 606(7913) 343-350), the clones assessed are likely, at least in part, to be competing with other clones that also possess a clonal advantage, something we now emphasize in the revised manuscript (Page 14).

Fig. 6e shows the stability of clonal production capacity of mature cells rather than the stability of clone size. Therefore, we replaced the referenced figure from **Fig. 6e** to **Fig. 6a–d** for this description.

8) Page 73, legend Extended Data Figure 2d: “Targeted genes are indicted by different colors”. That doesn’t seem to be the case. Different colors mark mutation types.

RESPONSE:

We thank the reviewer for carefully reviewing and pointing out this error for which we apologize. As this reviewer correctly points out, the coloring rather relates to mutation types. We have corrected this in a revised version of **Extended Data Fig2**.

9) Typo in Discussion, page 15: “Thus, by the time that a clone replenished from a single human HSC has become traceable through the mutational lineage fate mapping one is tracing the lineage contribution of an expanded clonal family of HSCs derived from a single HSC rather than a single HSC.”

RESPONSE:

We appreciate this reviewer’s careful reading and agree that this sentence was not clearly formulated. We have therefore revised this sentence (text page 15) to clarify:

“Thus, by the time that a clone replenished from a single human HSC has become traceable through mutational lineage fate mapping one is tracing the lineage contribution of an expanded clonal family of HSCs derived from a single HSC rather than the output of a single HSC.”